# Dynamic turnover of centromeres drives karyotype evolution in Drosophila

Ryan Bracewell, Kamalakar Chatla, Matthew J Nalley, Doris Bachtrog*

Department of Integrative Biology, University of California, Berkeley, Berkeley, United States

**Abstract** Centromeres are the basic unit for chromosome inheritance, but their evolutionary dynamics is poorly understood. We generate high-quality reference genomes for multiple *Drosophila obscura* group species to reconstruct karyotype evolution. All chromosomes in this lineage were ancestrally telocentric and the creation of metacentric chromosomes in some species was driven by de novo seeding of new centromeres at ancestrally gene-rich regions, independently of chromosomal rearrangements. The emergence of centromeres resulted in a drastic size increase due to repeat accumulation, and dozens of genes previously located in euchromatin are now embedded in pericentromeric heterochromatin. Metacentric chromosomes secondarily became telocentric in the *pseudoobscura* subgroup through centromere repositioning and a pericentric inversion. The former (peri)centric sequences left behind shrunk dramatically in size after their inactivation, yet contain remnants of their evolutionary past, including increased repeat-content and heterochromatic environment. Centromere movements are accompanied by rapid turnover of the major satellite DNA detected in (peri)centromeric regions.

DOI: https://doi.org/10.7554/eLife.49002.001

## Introduction

Centromeres are the chromosomal regions to which the spindle microtubules attach during each cell division to ensure disjunction of chromosomes. In many eukaryotes, centromeres consist of species-specific kilobase- to megabase-sized arrays of tandem repeats (*Willard, 1990*; *Birchler et al., 2009*; *Wang et al., 2009*; *Melters et al., 2013*). Centromeres are characterized by a distinct type of chromatin where histone H3 is replaced by a centromere-specific histone H3 variant (cenH3) (*Karpen and Allshire, 1997*; *Allshire and Karpen, 2008*), which interacts with other proteins to seed the kinetochore.

The specific chromosomal regions of centromere formation are typically highly repetitive and embedded in heterochromatin and are thus poorly studied in many species, and the mechanisms dictating centromere specification have remained elusive in most organisms. In budding yeast, characteristic sequences of ~120 bp fully specify centromere identity (*Hieter et al., 1985*). In organisms with complex repetitive centromeres, the site of kinetochore assembly is not strictly governed by primary DNA sequence but appears to be epigenetically determined (*Karpen and Allshire, 1997*; *Allshire and Karpen, 2008*). The key-determining factor in epigenetic centromere specification is the deposition of the centromere-specific histone H3 variant cenH3 (also known as CENP-A). Numerous factors have been identified that are required for cenH3 localization, but the mechanisms of centromere specification are not known in most species. Non-coding transcripts are associated with centromeres, and transcription might be linked to cenH3 chromatin assembly in some systems (*Chen et al., 2015*; *McNulty et al., 2017*; *Talbert and Henikoff, 2018*). The concept of epigenetic specification of the centromere is further supported by the existence of neo-centromeres, which form at ectopic chromosomal loci in many species (*du Sart et al., 1997*) and lack shared sequence features (*Burrack and Berman, 2012*). Several studies have found that centromeric satellite DNA

*For correspondence:
dbachtrog@berkeley.edu

Competing interests: The authors declare that no competing interests exist.

may form unusual secondary structures (termed non-B-form secondary structures, as opposed to the canonical B-form structure first described by Watson and Crick), and various types of non-B-form structures have been observed both in vitro and in vivo in centromeric DNA of various organisms, including single-stranded DNA, hairpins, R-loops and i-motifs (*Garavís et al., 2015*; *Kabeche et al., 2018*; *Kasinathan and Henikoff, 2018*). Thus, multiple factors including epigenetic marks, primary DNA sequence and secondary structure may act in concert to ensure the faithful formation of a centromere on each chromosome.

Centromeres are the basic unit for chromosome inheritance, but their evolution is highly dynamic. Both the centromeric histone protein (cenH3, which is called *Cid* in Drosophila), as well as the underlying DNA sequence that forms the centromere (that is, the centromeric satellite DNA) vary among species (*Henikoff et al., 2001*). Centromere repeats have been proposed to act as selfish genetic elements by driving non-Mendelian chromosome transmission during meiosis, thus prodding the rapid evolution of centromeric proteins to restore fair segregation (*Henikoff et al., 2001*; *Malik and Henikoff, 2009*).

In addition to their primary sequence, the position of centromeres along the chromosome varies widely as well and results in diverse karyotypes across species (*The Tree of Sex Consortium, 2014*). Centromeres can be found at the end of chromosomes or in the center (telocentric versus metacentric chromosomes). Changes in centromere position can be driven by chromosomal re-arrangements, moving an existing centromere to a new location, or by evolutionary centromere repositioning – a move of the centromere along the chromosome not accompanied by structural rearrangements (*Ferreri et al., 2005*; *Carbone et al., 2006*; *Rocchi et al., 2012*; *Schubert, 2018*; *Nishimura et al., 2019*). Thus, either pericentric inversions, or neo-centromere formation, that is the seeding of a novel centromere in a region previously not containing a centromere function, can drive karyotype evolution.

Studies of repetitive DNA, and the centromere in particular, are notoriously difficult. Most of our knowledge on centromere evolution is based on cytological studies, and comparison of chromosome morphology and mapping of repetitive DNA has provided a rich picture of the diversity of karyotypes. Karyotypes have been extensively investigated in Drosophila species, and the gene content of chromosome arms is conserved in flies (termed Muller elements; *Muller, 1940*; *Patterson and Stone, 1952*; *Ashburner, 1989*; *Whiting et al., 1989*). The ancestral karyotype of Drosophila consists of five large telocentric chromosomes (rods; termed Muller elements A-E) and the much smaller dot chromosome (Muller F), and novel karyotypes have originated repeatedly by chromosomal fusions and inversions. Flies in the *obscura* group harbor a diversity of karyotypes (*Figure 1*; *Sturtevant, 1936*; *Patterson and Stone, 1952*; *Buzzati-Traverso and Scossiroli, 1955*; *Powell, 1997*). The ancestral karyotype (found in flies of the *subobscura* group) consists of five ancestral telocentric rods and the dot chromosome, while species in the *obscura* and *affinis* group mainly contain metacentric chromosomes, and most chromosomes in the *pseudoobscura* group are telocentric. Both pericentric inversions and chromosomal fusions were invoked to explain this diversity in karyotypes (*Patterson and Stone, 1952*; *Buzzati-Traverso and Scossiroli, 1955*; *Segarra et al., 1995*; *Schaeffer et al., 2008*). Here, we use long-read sequencing techniques to assemble high-quality genomes for multiple species in the *obscura* group with different karyotypes, to reconstruct the evolutionary history of karyotypic evolution at the molecular level. Our assemblies recover entire chromosomes, including large fractions of repetitive DNA and pericentromere sequences and centromere-associated repeats. We uncover a dynamic history of centromere evolution in this species group, and identify centromeres likely being formed de novo in gene-rich regions, followed by dramatic size increases due to accumulation of repetitive DNA. In some species, these novel centromeres become obsolete secondarily, and centromere inactivation is accompanied by loss of repetitive DNA while maintaining a heterochromatic configuration. The transitions in karyotypes are associated with rapid turnover of centromere-associated repeats between species and suggest an important role for repetitive DNA in the evolution of reproductive isolation and the formation of new species.

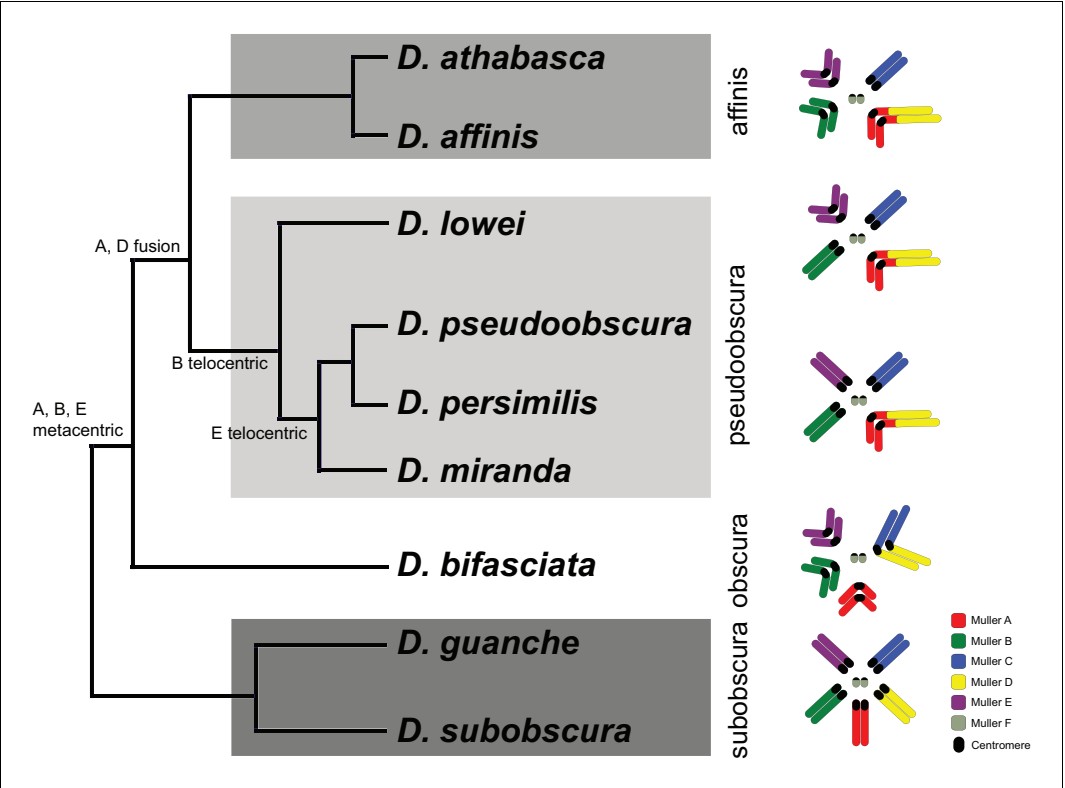

**Figure 1.** Phylogenetic relationships and karyotype evolution in the *D. obscura* group. *Drosophila subobscura* represents the ancestral karyotype condition consisting of five large and one small pair of telocentric chromosomes (termed Muller elements A-F). Phylogeny adapted from *Gao et al. (2007)*. Chromosomal fusions and movement of centromeres along the chromosomes has resulted in different karyotypes in different species groups (*Segarra et al., 1995*; *Schaeffer et al., 2008*). Indicated along the tree are transitions of chromosome morphology, and the different subgroups of the *obscura* species group are indicated by gray shading (the *subobscura*, *obscura*, *pseudoobscura*, and *affinis* subgroup). Muller elements are color coded, and centromeres are shown as black ovals.

DOI: https://doi.org/10.7554/eLife.49002.002

# Results

## Long-read sequencing allows assembly of entire chromosomes, including pericentromeric regions

Centromeres in many multicellular eukaryotes consist of highly repetitive satellite DNA, and are typically embedded in repeat-rich heterochromatin (that is, the pericentromeric heterochromatin; *Willard, 1990*; *Birchler et al., 2009*; *Wang et al., 2009*; *Melters et al., 2013*). Investigation of centromere biology has been greatly hampered by a lack of high-quality genome sequences (*Talbert et al., 2018*; *Chang et al., 2019*). Here, we obtain high-quality sequence assemblies of several Drosophila species in the *obscura* group, to reconstruct karyotype evolution at the molecular level.

We used a combination of PacBio and Nanopore sequencing and Hi-C scaffolding to generate highly contiguous genome assemblies of high quality for *D. pseudoobscura*, *D. athabasca*, *D. subobscura* and *D. lowei* (*Supplementary file 1*, *Supplementary file 2*, see Materials and methods and Appendix 1). We also included our recently published high-quality assembly of *D. miranda* to this analysis (*Mahajan et al., 2018*). For each species, we were able to generate chromosome-level assemblies that contain large stretches of repetitive DNA, including pericentromere and putative centromere sequence (*Figure 2*, *Figure 2—figure supplement 1*, *Figure 2—figure supplement 2*). In particular, euchromatic regions from each chromosome were typically assembled into just a few

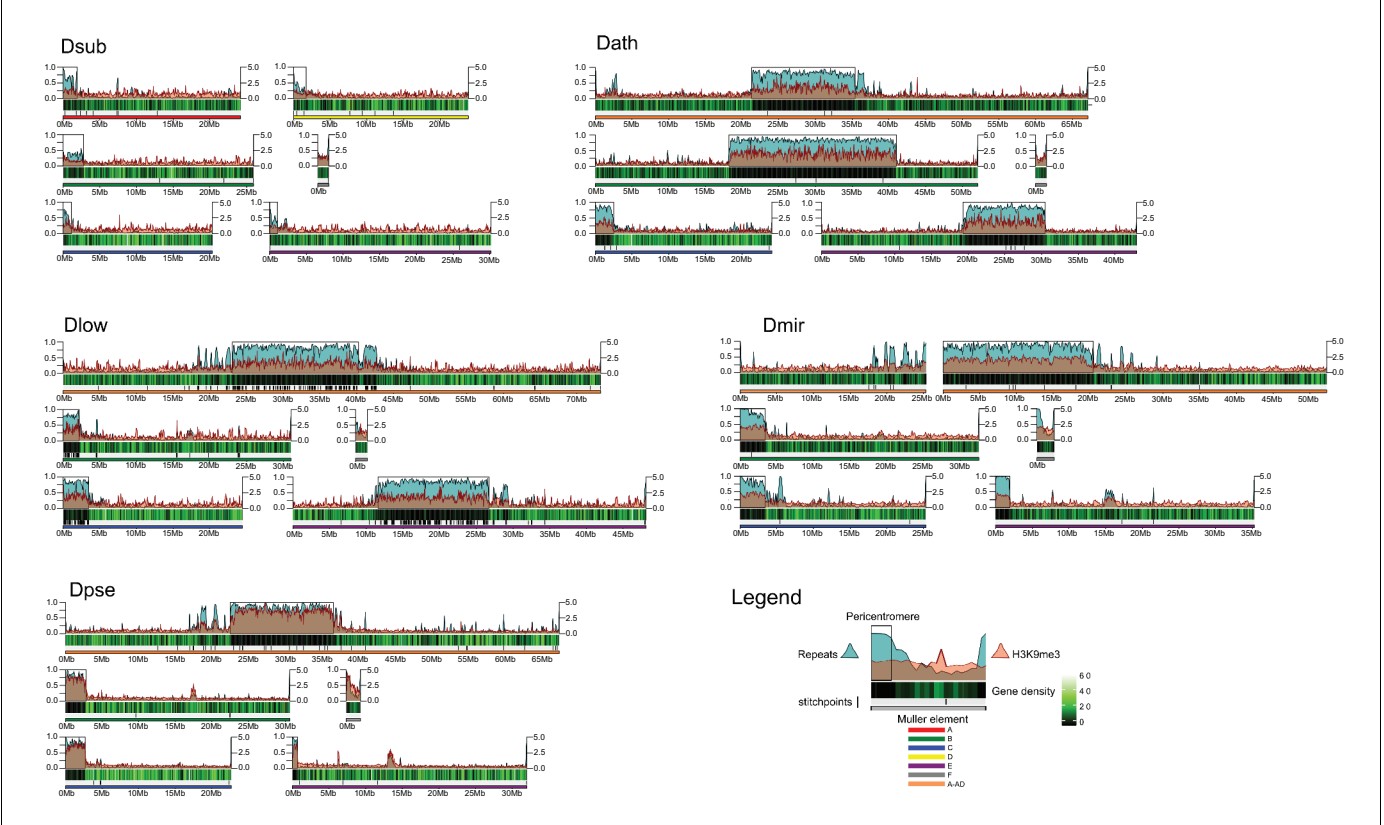

**Figure 2.** Genome organization in *Drosophila obscura* group flies. Shown here are the assembled chromosome sizes, scaffolding stitch points, gene density, repeat content (percentage of bases repeat-masked in 100 kb windows) and H3K9me3 enrichment (50 kb windows) across the genome assemblies of *D. subobscura*, *D. athabasca*, *D. lowei*, *D. pseudoobscura* and *D. miranda*. Muller elements are color coded, gene density is shown as a black to green heatmap (genes per 100 kb), H3K9me3 enrichment is shown in orange, and repeat density is shown in teal (note that H3K9me3 enrichment and repeat density are plotted semi-transparent). Scaffolding stich points are indicated as vertical lines.

DOI: https://doi.org/10.7554/eLife.49002.003

The following figure supplements are available for figure 2:

**Figure supplement 1.** Illumina coverage over assembled chromosomes in reference genome assemblies of *D. subobscura*, *D pseudoobscura*, *D. lowei* and *D. athabasca*.

DOI: https://doi.org/10.7554/eLife.49002.004

**Figure supplement 2.** Nanopore or PacBio coverage over reference genome assemblies of *D. subobscura*, *D pseudoobscura*, *D. lowei* and *D. athabasca*.

DOI: https://doi.org/10.7554/eLife.49002.005

**Figure supplement 3.** Hi-C association heatmaps of genome assemblies for *D. subobscura*, *D pseudoobscura*, *D. lowei* and *D. athabasca*.

DOI: https://doi.org/10.7554/eLife.49002.006

contigs (*Table 1*, *Figure 2*), and Hi-C scaffolding allowed for the orientation and placement of these contigs into chromosome-level assemblies (*Figure 2—figure supplement 3*). BUSCO (Benchmarking Universal Single-Copy Orthologs) scores reveal that our genomes are highly complete (*Supplementary file 3*) and match or exceed other recently published high quality Drosophila genomes (*Mahajan et al., 2018*; *Puerma et al., 2018*; *Puerma et al., 2018*; *Karageorgiou et al., 2019*). Identification of *D. melanogaster* orthologs in our genome assemblies allowed us to unambiguously assign each chromosome to a Muller element (*Figure 3A*) and further confirm the high level of gene content conservation across chromosomes in Drosophila (*Muller, 1940*; *Patterson and Stone, 1952*; *Bhutkar et al., 2008*). Our chromosome-level genome assemblies are in agreement with the karyotypes based on cytology (*Buzzati-Traverso and Scossiroli, 1955*; see *Figures 1–3*). Chromatin interaction maps display high levels of associations within chromosome arms and lower levels between (*Lieberman-Aiden et al., 2009*), and our Hi-C scaffolding maps reconstruct

**Table 1.** Total length (bp) of each assembled Muller element in each species, the number of contigs, and estimated length (Mb) of pericentromere sequence.

| Species | Muller A | Muller D | Muller A-AD | Muller B | Muller C | Muller E | Muller F | Total |
|---|---|---|---|---|---|---|---|---|
| *D. subobscura* | | | | | | | | |
| chromosome (bp) | 24,182,865 | 23,815,339 | n/a | 25,941,769 | 20,343,353 | 30,159,154 | 1,505,893 | 125,948,373 |
| contigs | 9 | 8 | n/a | 5 | 1 | 3 | 4 | 30 |
| pericentromere (Mb) | 1.9 | 1.7 | n/a | 2.8 | 1.1 | 1.1 | n/a | 8.6 |
| *D. athabasca* | | | | | | | | |
| chromosome (bp) | n/a | n/a | 67,112,822 | 52,101,127 | 24,053,775 | 42,973,490 | 1,524,173 | 187,765,387 |
| contigs | n/a | n/a | 4 | 4 | 6 | 7 | 1 | 22 |
| pericentromere (Mb) | n/a | n/a | 14.1 | 22.8 | 2.5 | 11.2 | n/a | 50.6 |
| *D. lowei* | | | | | | | | |
| chromosome (bp) | n/a | n/a | 73,251,623 | 31,032,897 | 24,430,087 | 48,132,706 | 1,606,711 | 178,454,024 |
| contigs | n/a | n/a | 190 | 42 | 47 | 152 | 1 | 432 |
| pericentromere (Mb) | n/a | n/a | 17.2 | 2.2 | 3.5 | 15.1 | n/a | 38 |
| *D. miranda* | | | | | | | | |
| chromosome (bp) | n/a | n/a | 77,621,844 | 32,539,841 | 25,306,191 | 35,263,383 | 2,366,016 | 173,097,275 |
| contigs | n/a | n/a | 18 | 2 | 3 | 3 | 1 | 27 |
| pericentromere (Mb) | n/a | n/a | 20.5 | 3.4 | 3.4 | 2 | n/a | 29.3 |
| *D. pseudoobscura* | | | | | | | | |
| chromosome (bp) | n/a | n/a | 67,434,674 | 30,637,803 | 22,641,560 | 32,023,297 | 1,941,385 | 154,678,719 |
| contigs | n/a | n/a | 37 | 6 | 5 | 5 | 1 | 54 |
| pericentromere (Mb) | n/a | n/a | 14.1 | 2.8 | 2.7 | 0.7 | n/a | 20.3 |

DOI: https://doi.org/10.7554/eLife.49002.016

five large telocentric chromosomes in *D. subobscura*, three large metacentric and two telocentric chromosomes in *D. athabasca*, two large metacentric and three telocentric chromosomes in *D. lowei* and four large telocentric and one metacentric chromosome in *D. pseudoobscura* and *D. miranda* (*Figure 2—figure supplement 3*).

We used transcriptome data for several species to aid in genome annotation (*Supplementary file 1*, *Supplementary file 4*). Our annotations identified 12,714 protein-coding genes in *D. subobscura*, 13,665 in *D. athabasca*, 14,547 in *D. lowei*, and 14,334 in *D. pseudoobscura*. The number of annotated genes in *D. subobscura* is very similar to the 13,317 protein-coding genes in another assembly of *D. subobscura* (*Karageorgiou et al., 2019*) and the 13,453 genes of its close relative *D. guanche* (*Puerma et al., 2018*). Further, the number of annotated genes in *D. pseudoobscura* and *D. lowei* (*Supplementary file 4*) are similar to the number in the current version of *D. pseudoobscura* (14,574 genes in Dpse_3.0). Currently, no high-quality genome assembly or annotation exists for a species that is closely related to *D. athabasca* (*affinis* subgroup), but our *D. athabasca* annotation is largely in line with the other species from the *obscura* group. We generated de novo repeat libraries for each species and concatenated them with a published *Drosophila* repeat library (*Bao et al., 2015*) to mask repeats in each genome. Our assemblies contained large amounts of repetitive DNA; we repeat-masked a total of 10.2 Mb, 54.4 Mb, 46.5 Mb, and 28.3 Mb from the assembled chromosomes of *D. subobscura*, *D. athabasca*, *D. lowei* and *D. pseudoobscura*, respectively (*Supplementary file 5*).

As expected, gene and repeat density are variable across all chromosomes in all species and show opposite patterns (*Figure 2*). We see sharp decreases in gene density near one end at telocentric chromosomes and near the center of metacentric chromosomes, while repeat density peaks in gene poor regions (*Figure 2*). Thus, our assemblies contain substantial amounts of (peri)centromeric repetitive DNA. We used repeat-enrichment to define the boundaries of euchromatic chromosome arms versus pericentromeric repeats for each species using a cut-off of 20% (*D. subobscura*) or 40%

(*D. athabasca*, *D. pseudoobscura*, *D. miranda* and *D. lowei*) repeat-masked DNA across 100 kb sliding windows away from the centromere (*Figure 2*). A lower cut-off was used for *D. subobscura*, as this species has a very small, compact and repeat-poor genome (see also *Karageorgiou et al., 2019*).

The sizes of (peri)centromere regions recovered for each chromosome and species vary considerably (*Table 1*). In most species, the identified pericentromeric regions are almost entirely consisting of masked repetitive DNA (*Figure 2*, *Supplementary file 5*), but pericentromeric regions in the *D. subobscura* assembly tend to be less repetitive (mean = 45.5%, *Figure 2*). The transition from repetitive pericentromere to euchromatic chromosome arms is also much less abrupt in this species (*Figure 2*). Genome-wide alignments between our *D. subobscura* assembly and a recently published high-quality genome confirm our inference of pericentromeric DNA (*Puerma et al., 2018*; *Karageorgiou et al., 2019*) and verify the orientation of the chromosomes (*Figure 3—figure supplement 1*). Further, comparison with another genome assembly in the *obscura* group (*D. guanche*; *Figure 3—figure supplement 1*) verifies known large chromosomal inversions between species and/or strains (*Orengo et al., 2019*) and provides further evidence of the quality of our assemblies.

We generated genome-wide H3K9me3 profiles for several species, which is a histone modification typical of heterochromatin (*Elgin and Reuter, 2013*). Indeed, we see significant enrichment of heterochromatin (as measured by H3K9me3) at pericentric regions (*Figure 2*). Thus, combined patterns of gene density, repeat content, and heterochromatin enrichment together with comparisons to published genomes confirm that our high-quality genome assemblies have allowed us to assemble large stretches of the pericentromere and putative centromeric regions in the various Drosophila species.

## Karyotype evolution in the *obscura* species group

The evolutionary relationships in this species group suggest that the ancestor of the *obscura* species group had the basic Drosophila karyotype consisting of five large telocentric rods and the dot chromosome (*Patterson and Stone, 1952*; *Buzzati-Traverso and Scossiroli, 1955*; *Gao et al., 2007*). Flies in the *subobscura* subgroup (which includes *Drosophila subobscura*, and its close relative *D. guanche*) harbor this ancestral chromosome configuration (*Patterson and Stone, 1952*; *Powell, 1997*), and whole genome alignments between these two species show long tracks of collinearity (*Figure 3—figure supplement 1*). In species of the *obscura* subgroup, Muller element A (which is the X chromosome in Drosophila), B and E have become metacentric (and in several members of this subgroup, element C and D have been fused; *Gao et al., 1949*). In an ancestor of the *pseudoobscura* and *affinis* subgroups, Muller element A and D fused. Previous research suggested that the fusion between Muller A and D involved two telocentric chromosomes, followed by a pericentric inversion translocating genetic material from Muller A to Muller D (*Segarra and Aguadé, 1992*; *Segarra et al., 1995*; note that we refer to these Muller elements as Muller A-AD). Phylogenetic considerations and genomic analysis, however, make a scenario of a fusion between an telocentric element D and a metacentric Muller element A more likely (see below; and also *Schaeffer et al., 2008*). Species in the *affinis* subgroup, which include *D. athabasca* and *D. affinis*, have maintained this karyotype where Muller A-AD are fused, and Muller B and E are metacentric. In some species of the *pseudoobscura* subgroup (that is, *D. miranda*, *D. pseudoobscura* and *D. persimilis*), however, both Muller B and E are telocentric, and parsimonious reconstruction suggests that these telocentric chromosomes evolved from a metacentric ancestor. Karyotyping in the little investigated species *D. lowei*, which is an outgroup to *D. miranda*, *D. pseudoobscura* and *D. persimilis*, suggests that only one of these autosomes is telocentric, while the other is still metacentric (*Heed et al., 1969*). Thus, the *obscura* group of Drosophila provides an exciting opportunity to study the evolution of vastly different karyotypes: All chromosomes in this group were ancestrally telocentric, three Muller elements became metacentric and two of these metacentric chromosomes secondarily reverted back to telocentric chromosomes.

Our high-quality genome assemblies recapitulate these large-scale changes in chromosome morphology, and comparison of gene content with *D. melanogaster* chromosomes allows us to unambiguously establish chromosome arm homologies (*Figure 3A*). Our assembly recovers five telocentric rods in *D. subobscura*, each flanked by repeat-rich pericentromeric DNA on one end. *D. athabasca*, in contrast, has three metacentric chromosomes, and our chromosome-level assembly recovers the (peri)centromeric regions in the center of these chromosomes (see *Figure 2*). We

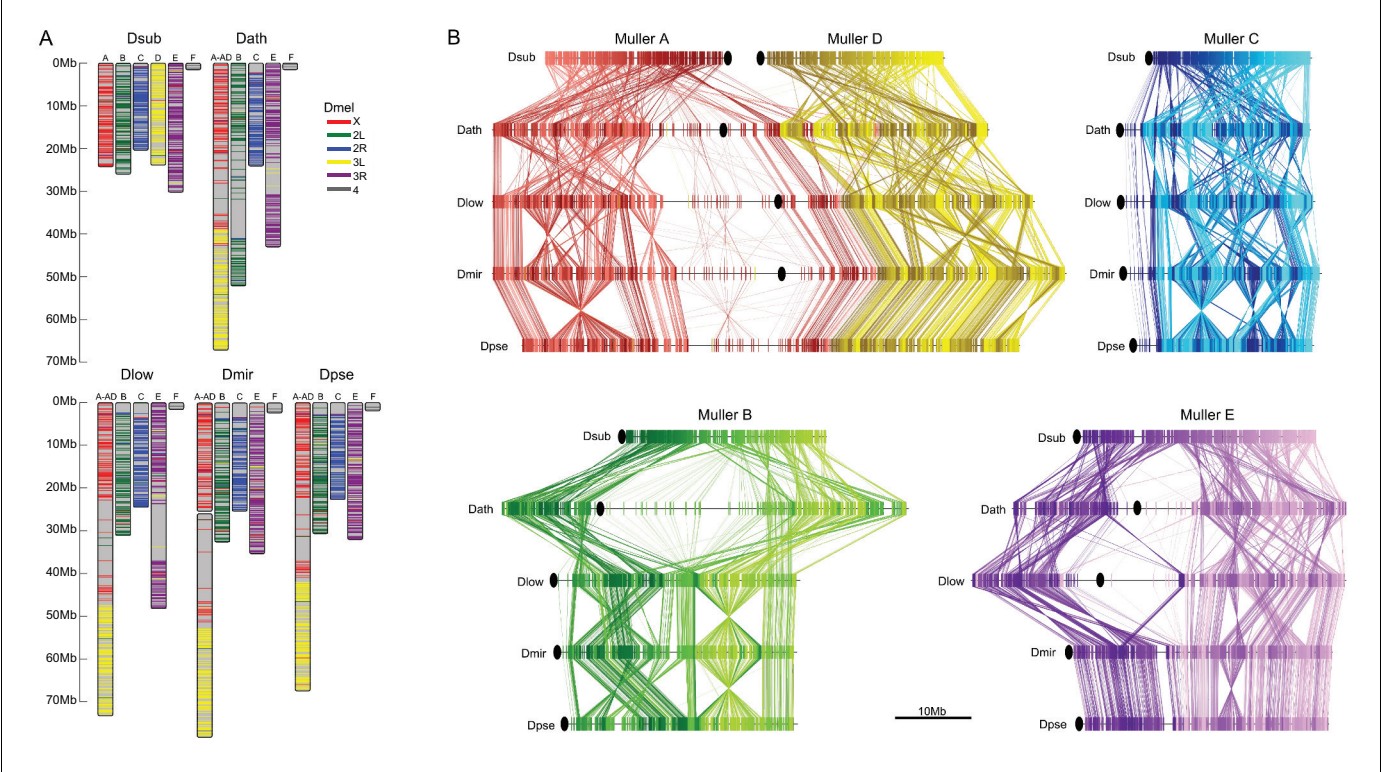

**Figure 3.** Chromosome synteny and evolution. (**A**) Conservation of Muller elements in the Drosophila genus. Orthologous single copy *Drosophila melanogaster* (Dmel) BUSCOs plotted on reference genome assemblies. Muller elements are color-coded based on *D. melanogaster*. (**B**) Comparisons of synteny between our genome assemblies. Muller elements are color-coded based on the *D. subobscura* genome. Each line represents a protein-coding gene. Ovals denote the location of the putative centromere (based on the location of centromere-associated satellite sequences, see *Figure 4*).
DOI: https://doi.org/10.7554/eLife.49002.007

The following figure supplements are available for figure 3:

**Figure supplement 1.** Whole genome alignments (MUMmer) of our *Drosophila subobscura* genome assembly (strain 14011–0131.10) with (**A**) *D. subobscura* strain *ch-cu* and (**B**) *D. guanche*.
DOI: https://doi.org/10.7554/eLife.49002.008

**Figure supplement 2.** Whole genome alignments (MUMmer) of draft *Drosophila bifasciata* genome contigs with (**A**) both Muller A and D of *D. subobscura* and (**B**) highlighting the Muller A centromere seed regions.
DOI: https://doi.org/10.7554/eLife.49002.009

assemble ~14 Mb of pericentric DNA on Muller A-AD, ~23 Mb on Muller B, and ~11 Mb on Muller E. In *D. lowei*, we assemble two telocentric large rods, and two metacentric chromosomes (Muller A-AD and E), and our assembly contains about 17 Mb of pericentric DNA on Muller A-AD and 15 Mb on Muller E (*Table 1*). *D. miranda* and *D. pseudoobscura*, in contrast, contain only a single metacentric chromosome (Muller A-AD) with 21 Mb and 14 Mb of assembled pericentromere sequence.

## Identification of centromere-associated repeats

Centromeres of many eukaryotes are associated with large stretches of satellite DNA, but simple satellites are typically missing from most assemblies. However, long-read sequencing technology has allowed resolving the detailed structure of complex satellites in *D. melanogaster* (*Khost et al., 2017*; *Chang et al., 2019*), and probing of high quality whole-genome assemblies based on long-read data has allowed for the recovery of putative centromere satellite motifs in *D. miranda* (*Mahajan et al., 2018*). In addition, given the extremely high repeat abundance of centromere repeats (often >10,000 copies per chromosome), the most abundant tandem repeat in a given genome is a prime candidate for the centromere repeat (*Melters et al., 2013*), and bioinformatic methods to identify high-copy tandem repeats from both raw sequencing reads or fragmented

assemblies have been applied to hundreds of species to study centromere evolution across animals and plants (*Melters et al., 2013*).

Here, we combine multiple complementary strategies to identify putative centromere-associated satellites in our Drosophila species (*Table 2*). We first used Tandem Repeat Finder (TRF; *Benson, 1999*) to identify satellite sequences in our assemblies that are enriched in putative centromeric regions (i.e., ends of telocentric chromosomes and scaffolding breakpoints in the middle of metacentric chromosomes). Additionally, we identify short simple satellites (<20 bp) directly from raw Illumina reads using k-Seek (*Wei et al., 2014*), and we also identify tandem repeats directly from long-read data (Nanopore and PacBio) using TideHunter (*Gao et al., 2019*). The latter two approaches provided analyses of satellite and repeat enrichment independent of our genome assemblies.

We previously annotated satellites in the *D. miranda* genome and recovered two repeats, a 21 bp motif and a 99 bp motif that were the most abundant tandem repeats (*Mahajan et al., 2018*). Both of these satellites were enriched at the putative centromere of each chromosome in our assembly and had sequence characteristics of typical centromere satellites (*Mahajan et al., 2018*). To identify putative centromere-associated repeats, we annotated satellites in our high-quality genomes and plotted the abundance of different length unique motifs (*Figure 4A,B*, *Figure 4—figure supplement 1*). Using this pipeline, our TRF analysis recovered the same 21 bp and 99 bp motif (or higher order variants of the two motifs) in *D. miranda*, with a higher order variant containing four 21 bp motifs (an 84 bp repeat) being the most common repeat unit (*Figure 4A*). We identify the same 21 bp motif in *D. pseudoobscura* (again with an 84-mer being the most common higher order variant, *Figure 4A*). Intriguingly, however, we fail to detect the 99 bp motif in *D. pseudoobscura*, both when probing for abundant satellites, and when searching the genome directly for the 99-mer. Similarly, we find a higher-order variant of the 21-mer in *D. lowei* (a 63-mer; *Figure 4A*), and only a distantly related 99-mer on Muller A-AD (mean = 18.1% sequence divergence). The 21-mer is enriched in the (peri)centric regions of most chromosomes in *D. pseudoobscura* and *D. lowei* (*Figure 4B*, *Figure 4—figure supplement 1*), except for Muller A-AD and the dot chromosomes in *D. pseudoobscura*. K-Seek analyses aimed at identifying abundant simple satellites failed to identify any highly enriched candidates (*Table 2*, *Figure 4—figure supplement 2*), consistent with previous work in *D. pseudoobscura* (*Wei et al., 2018*). Probing of PacBio and Nanopore long-reads with TideHunter was consistent with our TRF analyses and identified the 21-mer and higher-order variants in each of the three species (*Figure 4—figure supplement 3*). Further, we confirm the 99-mer in *D. miranda* and its absence in *D. pseudoobscura* and *D. lowei* (*Table 2*, *Figure 4—figure supplement 3*).

We used FISH hybridization to confirm the location of these satellites in the genomes of *D. miranda* and *D. pseudoobscura* (*Figure 4C*, *Supplementary file 6*, *Figure 4—figure supplement 4*). Indeed, we find both the 21-mer and the 99-mer enriched around the centromeres in *D. miranda* (*Figure 4C*). In contrast, FISH with the same probes in *D. pseudoobscura* only showed strong staining for the 21-mer around its centromeres but we failed to detect any signal for the 99-mer (*Figure 4C*). Consistent with our genome assembly, we failed to detect the 21-mer on Muller A-AD and the dot in *D. pseudoobscura* using FISH (*Figure 4C*). Thus, FISH experiments confirm our bioinformatics analysis, and strongly suggest that the composition of the (peri)centromeric repeat landscape has changed between *D. miranda* and *D. pseudoobscura*. In particular, we infer that the 21 bp repeat was present in an ancestor of the *pseudoobscura* subgroup with a (peri)centric distribution,

**Table 2.** Putative centromeric satellite lengths inferred from Tandem Repeat Finder (*Benson, 1999*), k-Seek (*Wei et al., 2014*), and TideHunter (*Gao et al., 2019*) for each Drosophila species. HOR = higher order repeat.

| | Tandem Repeat Finder | k-Seek | TideHunter |
|---|---|---|---|
| *D. subobscura* | no candidate | 12 bp | 12 bp, 107 bp |
| *D. athabasca* | 160 bp | 11 bp | 11 bp, 160 bp |
| *D. lowei* | 63 bp(HOR of 21 bp) | no candidate | 21 bp |
| *D. miranda* | 99 bp, 84 bp(HOR of 21 bp) | no candidate | 99 bp, 84 bp(HOR of 21 bp) |
| *D. pseudoobscura* | 84 bp(HOR of 21 bp) | no candidate | 168 bp(HOR of 21 bp) |

DOI: https://doi.org/10.7554/eLife.49002.017

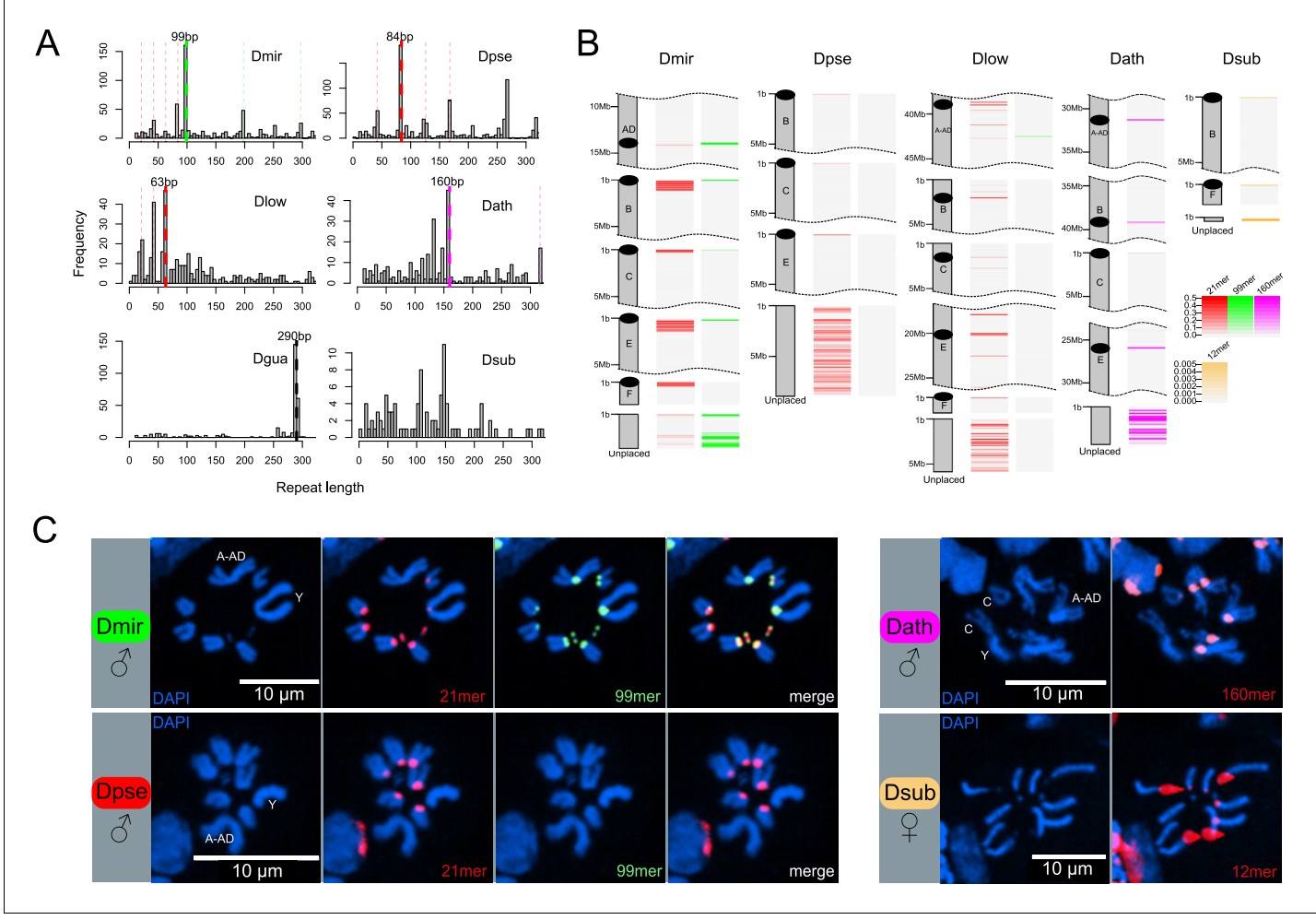

**Figure 4.** Identification of centromere-associated satellite sequences. (**A**) Histograms of most abundant satellites in assembled genomes. Repeat length refer to the size of the repeat unit. For each species apart from *D. subobscura*, a specific satellite (or higher-order variant of it as indicated by the same colors) is enriched. In *D. miranda*, a 99mer (in green) and four units of a unrelated 21mer (84 bp; in red) are the most abundant satellites, in *D. pseudoobscura*, four units of a similar 21mer (84 bp; in red) is most common, in *D. lowei*, three units of a similar 21mer (63 bp; in red) is most common, in *D. athabasca*, an unrelated 160mer (in pink) is the most common satellite, and in the *D. guanche* genome (*Puerma et al., 2018*), an unrelated 290mer (in black) is most common. No abundant satellite was identified in the assembled genome of *D. subobscura*. (**B**) Location of putative centromere-associated repeats (from panel A) in pericentromeric regions. In *D. suboobscura* a 12mer is highly enriched in raw sequencing reads. Shown is a 5 Mb fragment for each chromosome with the highest density of the satellite sequence (that is the putative centromere), and all unplaced scaffolds. (**C**) FISH hybridization confirms centromere location of identified satellites (same color coding as in A and B). Probes corresponding to the 21mer (Cy5; red) and 99mer (Cy3; green) were hybridized to both *D. miranda* and *D. pseudoobscura*; the 21mer showed a centromere location in both species, while the 99mer hybridized only to the centromeres of *D. miranda*. The 160mer (6FAM; red) localized to the centromeres of *D. athabasca*, and the 12mer (TYE665; red) to the centromeres of *D. subobscura*. Stronger hybridization signal supposedly correspond to higher repeat abundance at a particular genomic location.

DOI: https://doi.org/10.7554/eLife.49002.010

The following figure supplements are available for figure 4:

**Figure supplement 1.** Genomic distribution of inferred centromeric satellite sequences.
DOI: https://doi.org/10.7554/eLife.49002.011
**Figure supplement 2.** Short satellite DNAs in obscura group flies.
DOI: https://doi.org/10.7554/eLife.49002.012
**Figure supplement 3.** Identification of centromere-associated satellite sequences from Nanopore and PacBio reads.
DOI: https://doi.org/10.7554/eLife.49002.013
**Figure supplement 4.** Additional fluorescent in situ hybridization images.
DOI: https://doi.org/10.7554/eLife.49002.014

*Figure 4 continued on next page*

*Figure 4 continued*

**Figure supplement 5.** Nanopore and Illumina sequencing coverage over unplaced contigs (Contig_6 and Contig_7) harboring arrays of putative *D. subobscura* centromeric-associated satellite sequence.

DOI: https://doi.org/10.7554/eLife.49002.015

and that the 99 bp motif associated with (peri)centromeres in *D. miranda* was newly gained after their split.

In the *D. athabasca* genome, we identify a 160 bp motif as the most abundant repeat unit (*Figure 4A*), and this motif is enriched in the assembled (peri)centromere in both telocentric and metacentric chromosomes (*Figure 4B*, *Figure 4—figure supplement 1*). K-Seek and TideHunter analyses also identified a 11-mer (*Table 2*, *Figure 4—figure supplement 2*, *Figure 4—figure supplement 3*), which was less abundant in the genome assembly and in the raw PacBio reads than the 160 bp repeat (*Figure 4A*, *Figure 4—figure supplement 1*, *Figure 4—figure supplement 3*). FISH experiments again confirm our bioinformatics approach and demonstrate that the 160 bp satellite is localized to the centromeric region on all chromosomes except Muller F (*Figure 4C*). Lower intensity of fluorescence on Muller C suggests less satellite sequence on this chromosome (*Figure 4—figure supplement 4*).

In the *D. suboobscura* assembly, on the other hand, we fail to detect any strong enrichment of a particular satellite using TRF (*Figure 4A*). In its close relative *D. guanche*, in contrast, we clearly identify a known 290 bp centromere-associated satellite (*Bachmann et al., 2009*; *Puerma et al., 2018*). Thus, while *D. subobscura* has the most contiguous genome assembly of all species investigated (*Figure 2*), this suggests that we did not assemble its centromere-associated repeat. Computational identification and quantification of short tandemly repeating sequences using k-Seek revealed a 12 bp satellite that is highly enriched in raw Illumina whole genome sequencing reads of *D. subobscura* and yet appears largely absent from our assembly (or from *D. guanche* and all other species; *Figure 4—figure supplement 2*). Investigation of our Nanopore long reads also identified the 12 bp satellite and uncovered an additional 107 bp satellite in *D. subobscura* (*Table 2*, *Figure 4—figure supplement 3*). Inspection of unplaced scaffolds in *D. subobscura* reveals one short contig (14.8 kb in size) with extremely high Illumina and Nanopore read coverage (650x and 8500x in libraries sequenced to a genome wide average of 118x and 40x; *Figure 4—figure supplement 5*), and this contig contains two large arrays of the 12 bp satellite (one containing 278 copies and the other 250 copies). In addition, we identify this 12 bp array at the start of our assembly of Muller B (*Figure 4B*). Likewise, we find another unplaced short contig (14.5 kb in size) with moderately high Illumina sequencing coverage that contains the 107 bp satellite (*Figure 4—figure supplement 5*), and this satellite is enriched at the start of our assembly on Muller F (*Figure 4—figure supplement 1*). High read coverage suggests that the *D. subobscura* genome contains several Mb of the 12 bp and 107 bp satellite sequence, but that these simple satellite arrays are largely collapsed in our assemblies. FISH analysis confirms that the 12 bp repeat is enriched at all the centromeres of *D. suboboscura* chromosomes, except Muller F (and our genomic analysis suggests that the 107-mer is a good candidate for the centromere-associated repeat on Muller F). The intensity of fluorescence varied dramatically across chromosomes, and some chromosomes have exceptionally large foci that overwhelmed the visualization of other chromosomes (*Figure 4C*, *Figure 4—figure supplement 4*). This suggests that the size of the satellite array may differ drastically across different chromosomes.

Thus, our bioinformatics analysis identifies putative centromere-associated repeats in each species (*Table 2*), and FISH supports their location near the centromere. Importantly, however, our experiments do not allow us to identify the functional centromere. Indeed, initial FISH experiments in *D. melanogaster* suggested that the functional centromere in that species consist of simple satellites (*Lohe et al., 1993*; *Sun et al., 1997*; *Török et al., 2000*; *Jagannathan et al., 2017*; *Talbert et al., 2018*). However, recent high-resolution chromatin fiber imaging and mapping of the centromere protein instead indicate that island of retroelements, flanked by large arrays of satellites are the major functional components of centromeres in *D. melanogaster* (*Talbert et al., 2018*; *Chang et al., 2019*). It will be of great interest to perform similar high-resolution mapping studies in the *obscura* group, to determine the functional centromere and its evolution in this species group.

# Gene synteny reveals 'genomic accordions' associated with centromere gain and loss

Given the highly repetitive nature of centromeres, their embedment in repetitive pericentromeric DNA, and the fast turn-over of repeats and centromeric satellites between species, studying centromere evolution based on centromere (or pericentric) DNA itself is challenging. Instead, we used protein-coding genes inside or flanking pericentromeric DNA to obtain clues about the evolutionary history of centromere relocation and broad-scale patterns of karyotype evolution.

*Figure 3B* gives an overview of global syntenic relationships across the species investigated, based on the location of protein-coding genes; genes are assigned to Muller elements (and color-coded accordingly) based on their location in *D. suboobscura*. Consistent with previous studies within the Drosophila genus (*Bhutkar et al., 2008*), syntenic comparisons on all Muller elements reveal a rich history of intrachromosomal reshuffling of genes (*Figure 3B*). Muller C is telocentric in every species, but no major changes in overall chromosome size or repeat structure are found (*Figure 3B*, *Figure 2*). Likewise, Muller D fused to Muller A in more derived species of the *affinis* or *pseudoobscura* subgroups, but otherwise shows no major changes in size or composition. We detect a rearrangement in *D. athabasca* that moved some Muller A genes inside Muller D (*Figure 3B*), but the other three species with the Muller A and D fusion (*D. miranda*, *D. pseudoobscura* and *D. lowei*) do not show any reshuffling of genes between Muller A and D. This suggests that this re-arrangement occurred in the *D. athabasca* lineage, and argues against the proposed model of chromosome evolution in this species group (*Segarra and Aguadé, 1992*; *Segarra et al., 1995*). Rather than a fusion between a telocentric Muller A and D followed by a pericentric inversion translocating genetic material from Muller A to Muller D, synteny relationships reveal no evidence of a pericentric inversion in an ancestor of the *pseudoobscura* and *affinis* subgroups. In addition, analysis of a draft genome of *D. bifasciata* (a member of the *obscura* subgroup that lacks the Muller A-D fusion) further confirms that the centromere position on Muller A is conserved between *obscura*, *pseudoobscura* and *affinis* flies (*Figure 3—figure supplement 2*). Conservation of centromere position on Muller A corroborates that this chromosome was metacentric in an ancestor before the fusion between Muller A and D occurred.

In contrast, broad genomic comparisons reveal an accordion-like pattern of DNA gain and loss in chromosomes that have undergone a shift in centromere location (*Figure 3B*). The novel emergence of a centromere inside a chromosome arm results in a dramatic size increase in the genomic region now containing the centromere and pericentromere (that is, Muller A in *D. athabasca*, *D. lowei*, *D. miranda* and *D. pseudoobscura*; Muller B in *D. athabasca*; and Muller E in *D. athabasca* and *D. lowei*). Presumably, the invasion of repetitive elements in these newly formed 'neo-centromeres' has diluted their gene content and resulted in a dramatic increase of their size (see below). Intriguingly, once a centromere has been lost in the center of a chromosome, the genomic region associated with the now defunct 'paleo-centromere' dramatically decreases in size, and gene density recovers to a more typical level (that is Muller B in *D. lowei*, *D. miranda* and *D. pseudoobscura*; and Muller E in *D. miranda* and *D. pseudoobscura*).

## Emergence of new centromeres in gene-rich regions

To carefully reconstruct the molecular changes associated with the shift from an telocentric to a metacentric chromosome, we identified genes inside the pericentromere on Muller A, B, and E in species where these chromosomes are metacentric, and identify their homologs in *D. suboobscura*, the species with the ancestral karyotype. We focused on Muller A from *D. pseudoobscura*, and Muller B and E from *D. athabasca* (which have 42, 93, and 73 genes inside their pericentromere with a homolog in *D. suboobscura*, respectively), but the same conclusions are reached if we select metacentric chromosomes from other species (*Figure 5—figure supplement 1*). Note that gene annotations can be challenging in highly repetitive pericentromeric regions, likely leading to some misannotation. Therefore, we focused our analyses on orthologs identified through reciprocal best BLAST hits between genomes.

Intriguingly, the pericentromeric regions on each of the three metacentric chromosomes are largely syntenic with a small number of euchromatic regions in *D. suboobscura* that have a similar number and orientation of genes. This indicates that syntenic relationships are largely conserved and was utilized to define ancestral 'seed regions' of new pericentromeres (that is genomic regions in *D.*

*subobscura* that have become part of the pericentromere on the metacentric Muller A, B or E). In particular, we used reciprocal best BLAST hits between *D. pseudoobscura* or *D. athabasca* pericentromere genes and *D. subobscura* genes and found the edges of clustered hits (>2 sequential genes) in the *D. subobscura* genome to define the boundaries of pericentromere seed regions. Note that intrachromosomal rearrangements can break up ancestrally contiguous seed regions or secondarily incorporate additional fragments into a new pericentromere.

As suggested by our broad-scale synteny comparisons (*Figure 3B*), this more fine-scale look confirms that novel pericentromeres were seeded in previously gene-dense regions on Muller A, B and E, and dramatically extended in size by the accumulation of repetitive DNA (see *Figures 2* and *3*). The pericentromere on *D. pseudoobscura* Muller A is roughly 14.1 Mb (*Table 1*) and contains 42 genes with orthologs in *D. subobscura* (*Figure 5*). These genes are found in two 'seed regions' on Muller A in *D. subobscura*, whose combined size is 0.3 Mb. Comparison with *D. athabasca* identifies a third syntenic region in the *D. subobscura* genome that became incorporated into the pericentromere of the metacentric Muller A in *D. athabasca* (*Figure 5—figure supplement 1*). Similarly, the pericentromere of Muller B in *D. athabasca* is roughly 22.8 Mb, and we identify 93 pericentromere genes that are conserved in *D. subobscura* and found in two seed regions with a total size of 0.66 Mb. Finally, the pericentromere on *D. athabasca* Muller E is roughly 11.2 Mb with 73 conserved genes in two seed regions of *D. subobscura* with 0.88 Mb in size. Thus, the combined assembled pericentromers in the metacentric Muller A, B and E are ~48.1 Mb (*Table 1*), while the orthologous regions in *D. subobscura* are only ~1.81 Mb in size (i.e. a total increase over 27-fold). Each of the 'seed' regions has thus expanded dramatically and this expansion appears driven by the accumulation of repetitive sequences (*Figure 2* and below).

## Loss of metacentric location of centromeres

Intriguingly, in some species of the *pseudoobscura* subgroup, the metacentric autosomes secondarily reverted back to telocentric chromosomes (Muller B in all *pseudoobscura* species and Muller E in *D. miranda* and *D. pseudoobscura*). Comparisons of gene content between species reveal that the majority of the genes previously located in the metacentric pericentromere remain on the chromosome arm after this shift in centromere position happened (*Figure 3B*, *Figure 5*). That is, the centromere was moved while leaving behind the majority of genes in the pericentromeric region. Syntenic relationships remain mostly intact during this transition, and formerly pericentric genes in *D. athabasca* are found in two genomic segments each on both Muller B and E of *D. pseudoobscura* (*Figure 5*). However, the size of these former pericentromeric regions (i.e. paleocentromeres) has decreased substantially, through massive purging of intergenic DNA (mainly repetitive elements) while maintaining their gene complement. Specifically, while the pericentromere of Muller B in *D. athabasca* is 22.8 Mb, most homologous genes in *D. pseudoobscura* are located in a 0.8 Mb sized region in the center of Muller B (58 genes) and almost all remaining genes are found in the pericentromere (22 genes). Likewise, the homologous genes in the 11.2 Mb large pericentromere on *D. athabasca* Muller E are found in two segments along the homologous chromosome in *D. pseudoobscura* (they are roughly 0.96 Mb and 0.18 Mb in size and contain 58 and 12 genes). Similar islands are found in *D. miranda* at homologous positions on both Muller B and E (*Figure 2*; *Mahajan et al., 2018*). In *D. lowei*, the telocentric Muller B shows a similar pattern with the former metacentric pericentromere having drastically contracted in size (to 2.81 Mb). In contrast, Muller E is still metacentric in *D. lowei* (*Figure 2*). This indicates that the transition of a metacentric Muller B to a telocentric chromosome happened in an ancestor of the *pseudoobscura* subgroup, while Muller E only became telocentric after the split from *D. lowei*.

As a mark of their evolutionary past, we find that the repeat content in paleocentromeres is still increased above background levels, despite their massive reduction in size. In *D. pseudoobscura*, for example, the repeat content in euchromatic DNA (that is, non-pericentromere) is ~6%; in the paleocentromeres,~18% of DNA is masked for repeats on Muller B and ~32% on Muller E (*Figure 2*; *Supplementary file 5*). Similar patterns of repeat enrichment are seen for *D. miranda* (*Figure 2*; *Supplementary file 5*). In addition, paleocentromere islands on both Muller elements show a clear spike in heterochromatin enrichment (as measured by H3K9me3 profiles, see *Figure 2*; *Mahajan et al., 2018*). Thus, despite having been purged of a large fraction of repeats, these 'paleocentromeres' still contain clear signatures of their former life as a pericentromeric region, with the younger paleocentromere (on Muller E) showing higher repeat content.

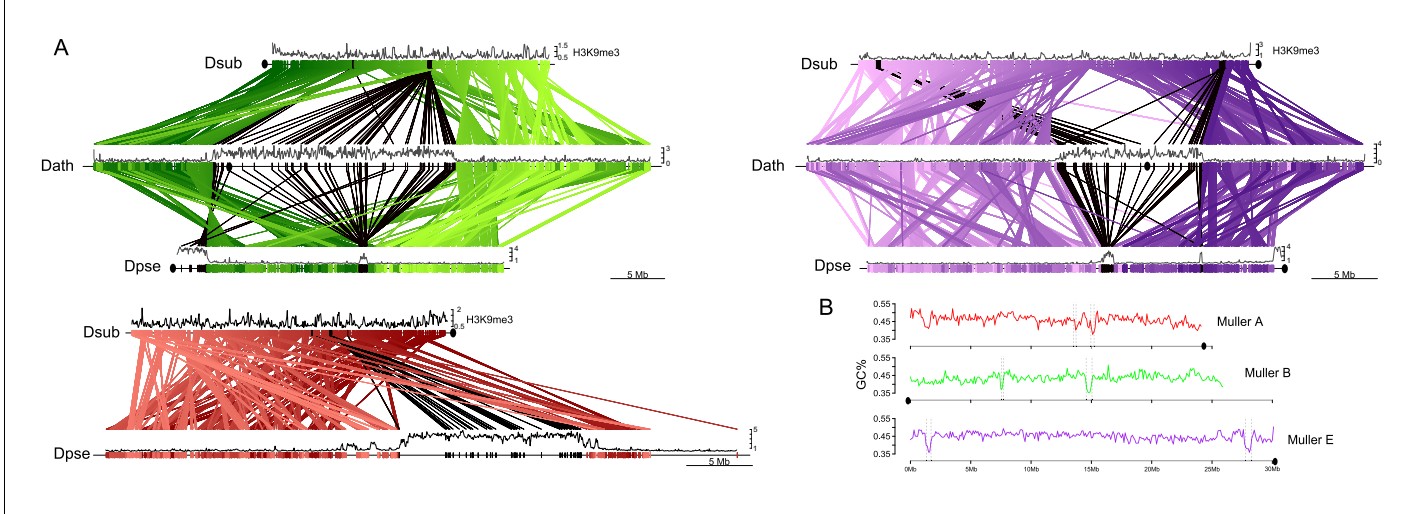

**Figure 5.** Emergence and loss of centromeres. (**A**) Shown are homologous genes between *D. subobscura* (telocentric), *D. athabasca* (metacentric) and *D. pseudoobscura* (metacentric and telocentric) with H3K9me3 enrichment plotted along Muller A (red), B (green) and E (purple) in 50 kb windows. Genes identified in the pericentromere of metacentric chromosomes are shown with black lines. Genes identified in pericentromeres of metacentric chromosomes can be traced to two 'seed regions' each on the telocentric chromosome of *D. subobscura*, and to paleocentromere regions in species that secondarily lost the metacentric centromere. (**B**) GC-content across *D. subobscura* Muller A, B and E. Seed regions have significantly lower GC-content compared to genomic background (***Supplementary file 7***).

DOI: https://doi.org/10.7554/eLife.49002.018

The following figure supplements are available for figure 5:

**Figure supplement 1.** Alignments of Muller A between *D. subobscura* (telocentric) with *D. athabasca* (metacentric) and H3K9me3 enrichment plotted in 50 kb windows above each chromosome.
DOI: https://doi.org/10.7554/eLife.49002.019

**Figure supplement 2.** GC-content, the percentage of bases repeat-masked, and number of genes, in 10 kb non-overlapping windows across Muller A, B and E.
DOI: https://doi.org/10.7554/eLife.49002.020

**Figure supplement 3.** GC-content of different functional categories in seed and non-seed regions of Muller A, B and E of *D. subobscura*.
DOI: https://doi.org/10.7554/eLife.49002.021

Phylogenetic dating allows us to roughly estimate when these evolutionary transitions occurred and suggests that the birth and death of a centromere can happen quickly on an evolutionary time scale. Molecular dating in this species group suggests that the *subobscura* subgroup split from the other species groups about 20MY ago (*Gao et al., 2007*). The *affinis* and *pseudoobscura* subgroups split from the *obscura* flies about 16MY ago and diverged from each other roughly 14MY ago (*Gao et al., 2007*). *D. lowei* diverged about 9MY ago, and *D. miranda* and *D. pseudoobscura* split roughly 3MY ago (*Gao et al., 2007*). These rough time estimates suggest that Muller A, B and E all became metacentric 16-20MY ago. Muller A and D subsequently fused (14-16MY ago), and Muller B reverted to a telocentric chromosome 9-14MY ago, while Muller E shifted its centromere only 3-9MY ago.

## Centromere repositioning drives karyotype differences

What mutational events are creating the diversity of karyotypes across flies from the *obscura* subgroup? Centromeres can shift along the chromosome by either moving the existing centromere to a new location through structural rearrangements, or by de novo formation of a centromere at a new genomic position (***Figure 6A***). Pericentric inversions are typically invoked to explain transitions between metacentric and telocentric chromosomes in flies (***Muller, 1940***; ***Patterson and Stone, 1952***). In particular, pericentric inversions where one breakpoint is located within the pericentromere and the other in the euchromatic arm can easily move the centromere along the chromosome, and transform a metacentric chromosome to a telocentric one, and vice versa (***Figure 6A***). Alternatively, centromeres may emerge de novo in a previously euchromatic region by centromere repositioning

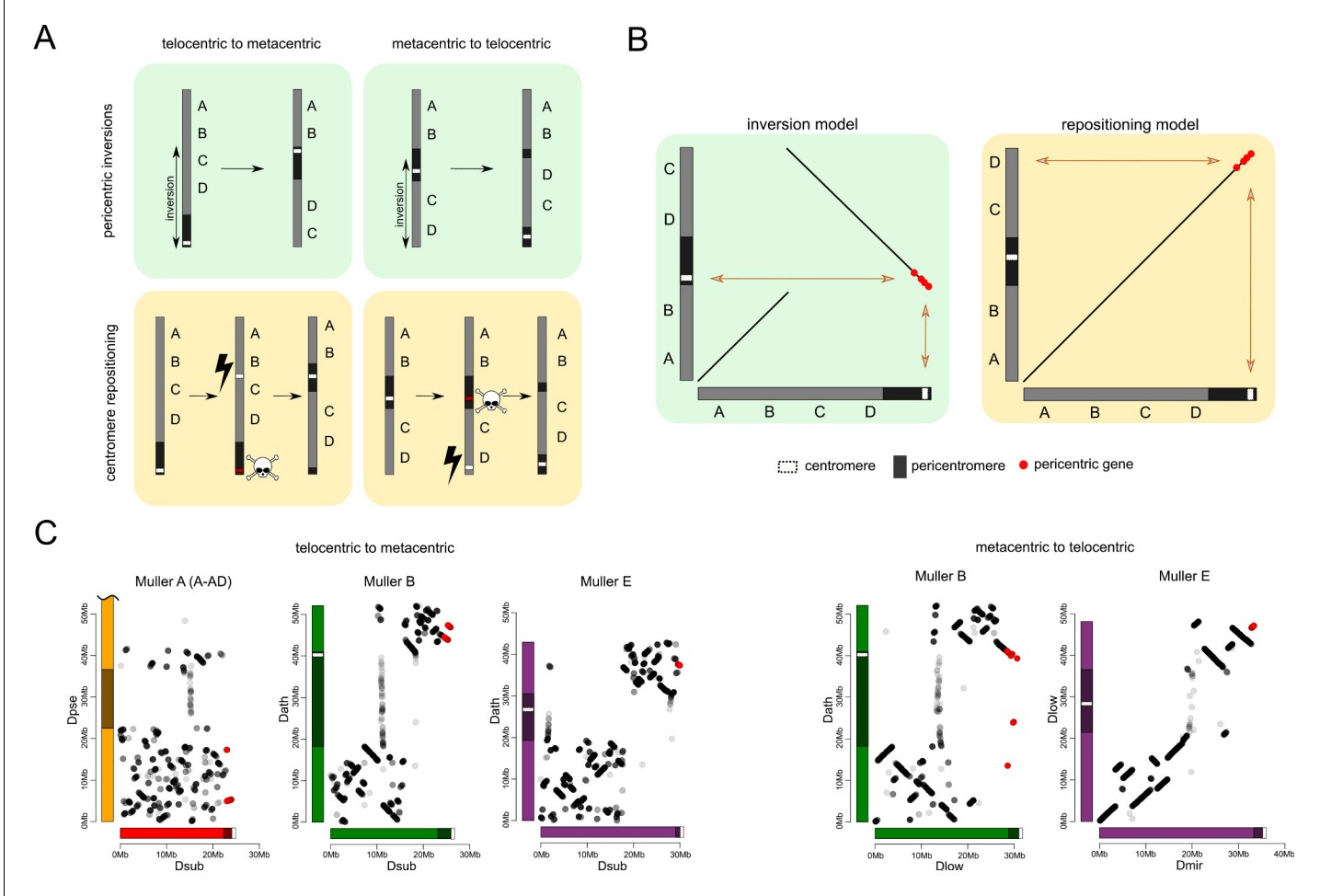

**Figure 6.** Karyotype and centromere evolution. (**A**) Models for transitions between metacentric and telocentric chromosomes, either invoking pericentric inversions (top), or centromere repositioning (bottom) via the birth of a new centromere (lightning bolt) and death of the old centromere (skull and crossbones). The pericentromere is indicated by darker shading, the centromere as a white rectangle. (**B**) The syntenic location of genes adjacent to the centromere can allow us to distinguish between a simple inversion model vs. centromere relocation. The genes closest to the centromere of the telocentric chromosome (30 genes in panel C) are shown by different shading. (**C**) Dot plots for homologous genes (semi-transparent points) between telocentric and metacentric Muller elements (orange: Muller A-AD; purple: Muller E; green: Muller B). In 4 out of 5 cases, pericentric genes in the telocentric species are found in the non-pericentric regions of the metacentric species. Only Muller B between *D. athabasca* and *pseudoobscura* group flies (*D. lowei* is pictured) shows that the same genes are pericentric in both species (and thus support a simple inversion model).
DOI: https://doi.org/10.7554/eLife.49002.022

(*Figure 6A*). This model involves the in situ creation of a neo-centromere without translocation of DNA, concordant with the inactivation of the ancestral centromere. The initial shift in the position of the centromere may be caused by epigenetic changes, resulting in the incorporation of the centromeric histone protein cenH3 at a novel position along the chromosome, and simultaneous silencing of the ancestral centromere (*Amor et al., 2004*; *Tolomeo et al., 2017*; *Schubert, 2018*).

The reconstruction of syntenic relationships between species with different karyotypes should allow us to infer the mutational mechanisms driving the movement of centromeres across the genome, yet the high frequency of inversions in Drosophila makes the exact reconstruction of karyotype evolution difficult. Also, the repetitive nature of centromeres, and their fast turnover between species makes centromere DNA itself unsuitable for distinguishing between models of centromere movement, but unique regions within or adjacent to pericentromeric heterochromatin can be used (*Figure 6B*). If pericentromeric inversions move the centromere along the chromosome, we expect that some genes are part of the pericentromere in both karyotypic configurations (*Figure 6B*). In contrast, if centromeres are seeded de novo at a new genomic location, we expect no overlap in

pericentromeric gene content (or genes immediately flanking the pericentromere) between species with different karyotypes (*Figure 6B*). Note that we cannot refute more complex structural rearrangements, such as subsequent peri- and paracentric inversions with breaks flanking the core centromere repeat on either side, where the pericentric inversion moves the centromere to a new chromosomal position, followed by a paracentric inversion moving formerly pericentromeric genes to another location (*Schubert, 2018*).

Genome-wide alignments between *D. subobscura* (or *D. guanche*) and *D. athabasca* or species from the *D. pseudoobscura* subgroup are consistent with centromere repositioning creating novel centromeres in the middle of Muller A, B and E from an ancestrally telocentric chromosome (*Figure 5*). In particular, all the homologs of genes present in the pericentromere region of all three metacentric Muller elements are located in euchromatic, gene-rich regions in *D. subobscura* (*Figure 5*). This is consistent with a model where the emergence of metacentric Muller A, B and E elements was due to de novo origination of a centromere in the middle of these chromosomes (but could also be caused by complex structural re-arrangements).

Close inspection of the genes adjacent to the centromeres on Muller A, B and E in *D. subobscura* further shows that their homologs are located nowhere near the (peri)centromere in species with metacentric chromosomes (*Figure 6C*). Instead, *D. subobscura* (peri)centromeric genes on Muller A are found in the middle of the long arm of Muller A in *D. pseudoobscura*, pericentromeric Muller B genes are located in the center of the short arm of Muller B in *D. athabasca*, and pericentromeric Muller E genes in *D. subobscura* are found in the center of the short arm of Muller E in *D. athabasca* (*Figure 6C*). As mentioned above, an additional genomic fragment that is adjacent to the centromere of *D. subobscura* became incorporated into the pericentromere of Muller A in *D. athabasca* (*Figure 5—figure supplement 1*), and pericentric genes in *D. subobscura* appear in the pericentromere of *D. athabasca* but not near the inferred centromere position (*Figure 5—figure supplement 1*). Thus, genomic comparisons do not support a simple model where pericentric inversions move the centromere from a telocentric location to the center of the chromosome, as previously proposed. However, we cannot exclude complex rearrangements (such as multiple nested inversions, chromosome breakage, etc.) contributing to centromere shifts in *obscura* flies.

In species of the *pseudoobscura* subgroup, Muller B and E have become telocentric secondarily. The shift of the centromere on Muller E from a central to a terminal location again shows no evidence for involvement of inversions. Homologs of genes found in the pericentromere of *D. athabasca* (or *D. lowei*) are all located along the euchromatic chromatin arm in both *D. miranda* and *D. pseudoobscura*, mega-bases away from the centromere on Muller E (*Figure 5*). Concordantly, the homologs of genes adjacent to the centromere on Muller E of *D. miranda* are found on the short arm of Muller E in *D. lowei*, nowhere close to the centromere in this species (*Figure 6C*).

In contrast, we find evidence for a pericentric inversion transforming a metacentric Muller B into a telocentric chromosome in the *pseudoobscura* subgroup. In particular, we find that the homologs of genes located within the pericentromere of *D. athabasca* are split in *pseudoobscura* group flies, with most of them remaining in the center of the chromosome (and forming the heterochromatic paleocentromere discussed above). Intriguingly, a second set of genes that is pericentric in *D. athabasca* is also located within the pericentromere of Muller B in *D. pseudoobscura* flies (*Figure 5*). In addition, genes that flank the telocentric centromere in *D. lowei* also border the metacentric centromere of *D. athabasca* (*Figure 6C*). Thus, these syntenic relationships suggest that a pericentric inversion was involved in the transition of a metacentric Muller B to a telocentric chromosome in the *pseudoobcura* group. However, the inferred inversion has left behind the majority of genes from the ancestral pericentromere, which constitute the heterochromatic paleocentromere in *pseudoobscura* flies described above. Thus, while our data do not allow us to rule out complex structural re-arrangements, we infer that simple centromere repositioning events underlie most centromere shifts observed in the *obscura* group, and the emergence of new karyotypes. However, pericentric inversions also contribute to karyotype evolution.

## Seed regions for neo-centromeres are AT-rich

What molecular characteristics drive the emergence of a new centromere in a previously euchromatic region? The genomic regions in *D. subobscura* that became incorporated into the pericentromeres in *D. athabasca* and the *pseudoobscura* subgroup (i.e. the pericentromere 'seed' regions) overall look like random, anonymous regions in *D. subobscura* (*Figure 2*, *Figure 5—figure*

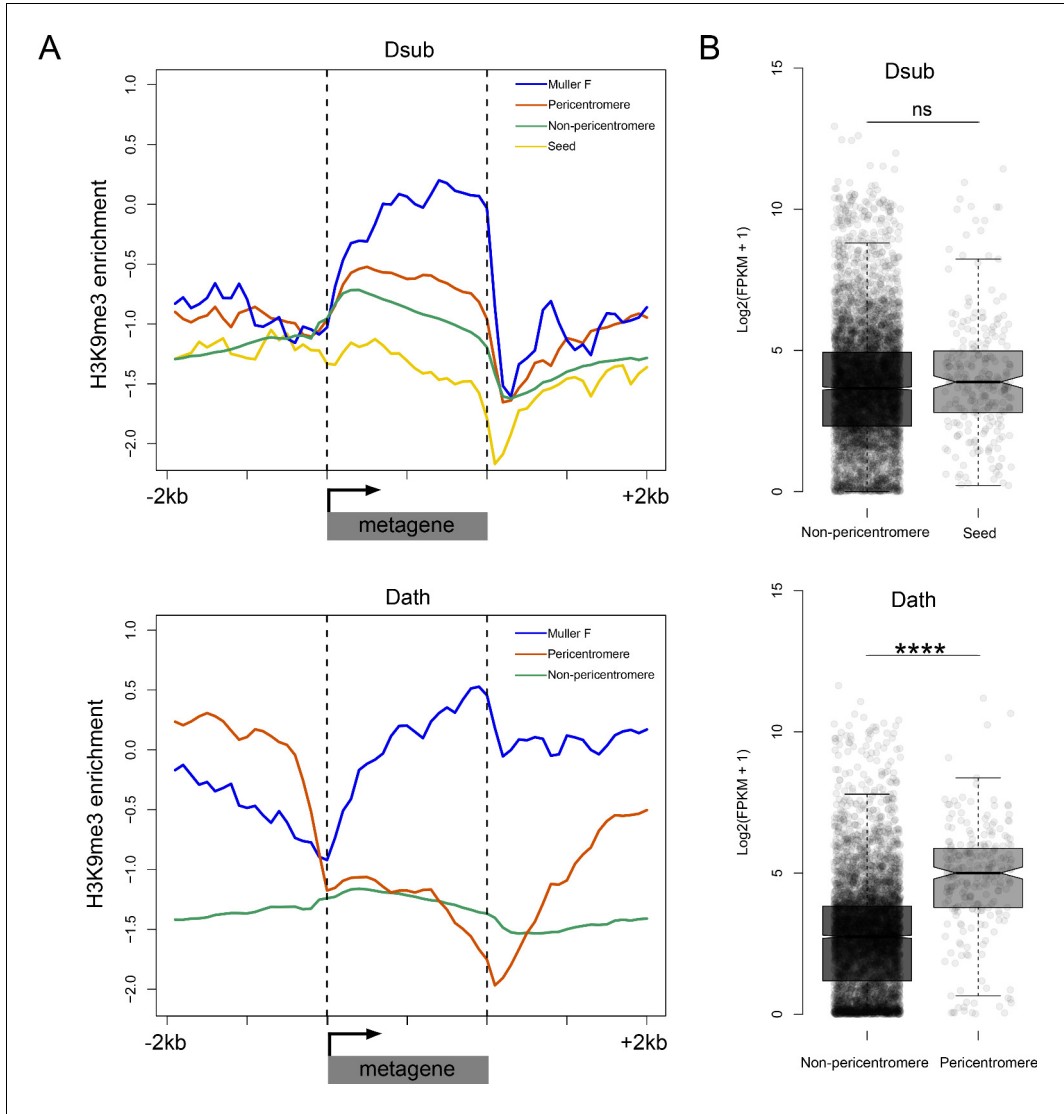

**Figure 7.** Functional consequences of becoming pericentromeric. (**A**) Metagene plots showing H3K9me3 enrichment for genes located in different parts of the genome in *D. subobscura* (top) and *D. athabasca* (bottom). (**B**) Patterns of gene expression for homologous genes in *D. subobscura* and *D. athabasca*, classified as whether they are part of the 'seed' region in *D. subobscura* that become part of the pericentromeric heterochromatin in *D. athabasca* or not. Expression patterns were not found to significantly differ between *D. subobscura* non-pericentromeric genes and seed genes, while seed orthologs located in the pericentromere of *D. athabasca* showed significantly higher expression than non-pericentromeric genes (Mann-Whitney U, p<0.0001).
DOI: https://doi.org/10.7554/eLife.49002.023

*supplement 2*). Both gene density and repeat density in these 'seed' regions appear within the bounds of background levels of gene and repeat density in euchromatic regions in *D. subobscura*. We do detect an overall significant increase in repeat content and gene content (*Supplementary file 7*); however, this appears to be driven largely by properties of individual seeds (e.g, the increased repetitiveness of the Muller E seed close to the pericentromere or the increased gene density in one of the Muller B seeds, *Figure 5—figure supplement 2*). The most unusual aspect of each of these seeds regions is that they are very AT-rich (*Figure 5B*; *Supplementary file 7*). To better understand what may drive this increase in AT content, we separately compared AT content in intergenic, genic and exon regions (*Figure 5—figure supplement 3*). Interestingly, we find that AT content is significantly increased for each functional category in seed regions, relative to

genomic background (*Supplementary file 7*). Thus, certain chromosomal regions may be predisposed to centromere formation, possibly because of their high AT content.

## Gene evolution in neo-centromere and paleo-centromere

In primates, newly evolved centromeres are notably deficient in genes, suggesting that centromere function is incompatible with gene activity (*Lomiento et al., 2008*). In contrast, the neo-centromeres in *D. athabasca* and flies of the *pseudoobscura* group appear to have been seeded in previously gene-rich regions on Muller A, B and D (*Figure 5*). For example, the genomic regions that became part of the centromere harbor a total of 245 genes annotated on Muller A, B and E in *D. athabasca.* The formation of the (peri)centromere is accompanied by a massive gain of repetitive DNA, and a change in the chromatin environment for genes in these regions, with a global increase in H3K9me3 enrichment (*Figure 2*). Thus, genes formerly located in euchromatic chromosome arms are now found in pericentromeric heterochromatin. In contrast to the major increase in repetitive DNA that we observed when a centromere is newly gained, we find that the former centromeres (or paleo-centromeres) in *D. miranda* and *D. pseudoobscura* are losing repeats, yet still maintain a heterochromatic environment (*Figure 2*).

To evaluate the functional significance of the changes in chromatin environment, we contrasted chromatin structure and expression levels of genes across species and genomic regions. We generated metagene plots of H3K9me3 enrichment for different categories of genes and their flanking sequences in *D. subobscura* and *D. athabasca*. As expected, genes located on the chromosome arms outside the pericentromeric region generally show no enrichment of H3K9me3 (*Figure 7A*). In contrast, the small dot chromosome of Drosophila is heterochromatic yet gene-rich, and genes on this chromosome have adopted a unique chromatin structure (*Riddle et al., 2012*). While there is a depletion of repressive chromatin marks at their transcription start site, H3K9me3 is strongly enriched over gene bodies (*Riddle et al., 2012*), and this unique chromatin signature is regulated by a molecular pathway unique to the dot chromosome. Genes in pericentric heterochromatin of *D. melanogaster*, on the other hand, have a distinct chromatin structure and do not show this enrichment of H3K9me3 at their gene bodies (*Riddle et al., 2012*). Metagene plots for genes located on the dot chromosome for *D. athabasca* and *D. suboobscura* are similar to those in *D. melanogaster* and show a depletion of H3K9me3 at their promoter region, but an enrichment over gene bodies (*Figure 7A*). Genes in the pericentromere of *D. subobscura* resemble pericentromeric genes in *D. melanogaster*, with rather uniform levels of H3K9me3 across the genes and flanking regions, at slightly higher levels than euchromatic genes (*Riddle et al., 2011*; *Riddle et al., 2012*). Genes in the neo-centromeres of *D. athabasca* show a unique chromatin signature: H3K9me3 levels are depleted at their transcription start site and across their gene body, yet H3K9me3 levels are strikingly elevated at flanking regions (*Figure 7A*). Thus, while pericentric genes in *D. athabasca* are embedded in repressive heterochromatin, silencing marks do not seem to spread across gene bodies. In line with a lack of repressive chromatin marks, we find no reduction in expression levels of genes within pericentromeric regions of metacentric chromosomes in *D. athabasca* relative to genes located in neighboring euchromatin (in fact, genes are expressed at significantly higher levels inside the pericentromere; *Figure 7B*). Levels of gene expression between homologous genes in *D. athabasca* and *D. suboobscura* ('seed' vs. pericentromere, *Figure 7B*), further demonstrate that pericentric genes in *D. athabasca* are not expressed at a lower level, despite being embedded in a repressive heterochromatic environment.

## Repeat evolution in pericentromeric regions

We show that newly formed (peri)centromeres dramatically increase in size, due to a massive gain in repetitive elements. To investigate what types of repeats are driving this size increase, and whether similar elements cause size expansion in different chromosomes and species, we categorized repeats from our assembled genomes into different repeat families and elements, using a curated TE library (*Hill and Betancourt, 2018*). We also performed an independent characterization of repetitive elements using de novo assembly of repeats with dnaPipeTE (*Goubert et al., 2015*). Overall, we find dramatic differences in repeat density and composition between species (*Table 3*, *Figure 8*, *Supplementary file 8*). As mentioned, *D. subobscura* chromosomes have by far the lowest fraction of bases masked for TEs (6%) and only show slight enrichment near the assembled pericentromeres

**Table 3.** Transposable elements in the *D. obscura* species group.

| Species | TE | Total bp masked | % of genome masked |
|---|---|---|---|
| *D. subobscura* | | | |
| | total TE's | 7,572,806 | 6.0% |
| | Dpse_Gypsy_6 | 1,319,782 | 1.0% |
| | CR1-1_DPer | 449,484 | 0.4% |
| | Gypsy8-I_Dpse | 424,330 | 0.3% |
| | LOA-1_DPer | 331,166 | 0.3% |
| | T213_X.Unknown | 323,480 | 0.3% |
| *D. athabasca* | | | |
| | total TE's | 42,382,296 | 22.6% |
| | Daff_Jockey_18 | 3,779,133 | 2.0% |
| | CR1-1_DPer | 2,938,986 | 1.6% |
| | T32_LTR | 1,958,710 | 1.0% |
| | LOA-1_DPer | 1,729,016 | 0.9% |
| | LOA-2_DPer | 1,683,407 | 0.9% |
| *D. lowei* | | | |
| | total TE's | 45,307,006 | 25.4% |
| | HelitronN-1_DPe | 2,764,830 | 1.5% |
| | CR1-1_DPer | 2,564,981 | 1.4% |
| | LOA-3_DPer | 1,421,120 | 0.8% |
| | LOA-2_DPer | 1,019,792 | 0.6% |
| | BEL-3_DPer-I | 965,011 | 0.5% |
| *D. pseudoobscura* | | | |
| | total TE's | 29,907,407 | 19.3% |
| | CR1-1_DPer | 2,082,339 | 1.3% |
| | HelitronN-1_DPe | 1,962,822 | 1.3% |
| | LOA-3_DPer | 1,024,994 | 0.7% |
| | T154_X.Unknown | 799,007 | 0.5% |
| | LOA-2_DPer | 726,556 | 0.5% |
| *D. miranda* | | | |
| | total TE's | 42,680,234 | 24.7% |
| | HelitronN-1_DPe | 3,834,913 | 2.2% |
| | CR1-1_DPer | 3,148,617 | 1.8% |
| | Gypsy18-I_Dpse | 2,261,130 | 1.3% |
| | LOA-3_DPer | 1,557,989 | 0.9% |
| | LOA-2_DPer | 1,208,941 | 0.7% |

DOI: https://doi.org/10.7554/eLife.49002.027

(*Table 3*, *Figure 8*, *Figure 8—figure supplement 1*). Genomic repeat content is substantially higher for the other species, ranging from 19–25% (*Table 3*) and these patterns were qualitatively similar in our dnaPipeTE analyses (*Figure 8—figure supplement 2*). Repeat composition also differs among taxonomic groups. A retrotransposable element belonging to the *Gypsy* family (Dpse_Gypsy_6) is the most prevalent TE in *D. subobscura*, and accounts for 17% of TE-derived DNA in that species (*Table 3*, *Supplementary file 9*). The most common repeat in *D. athabasca* belongs to the Jockey family, with 9% of all TEs derived from the Daff_Jockey_18 element (*Table 3*). In species of the *pseudoobscura* subgroup, on the other hand, a LINE repeat belonging to the CR1 family (CR1-1_DPer)

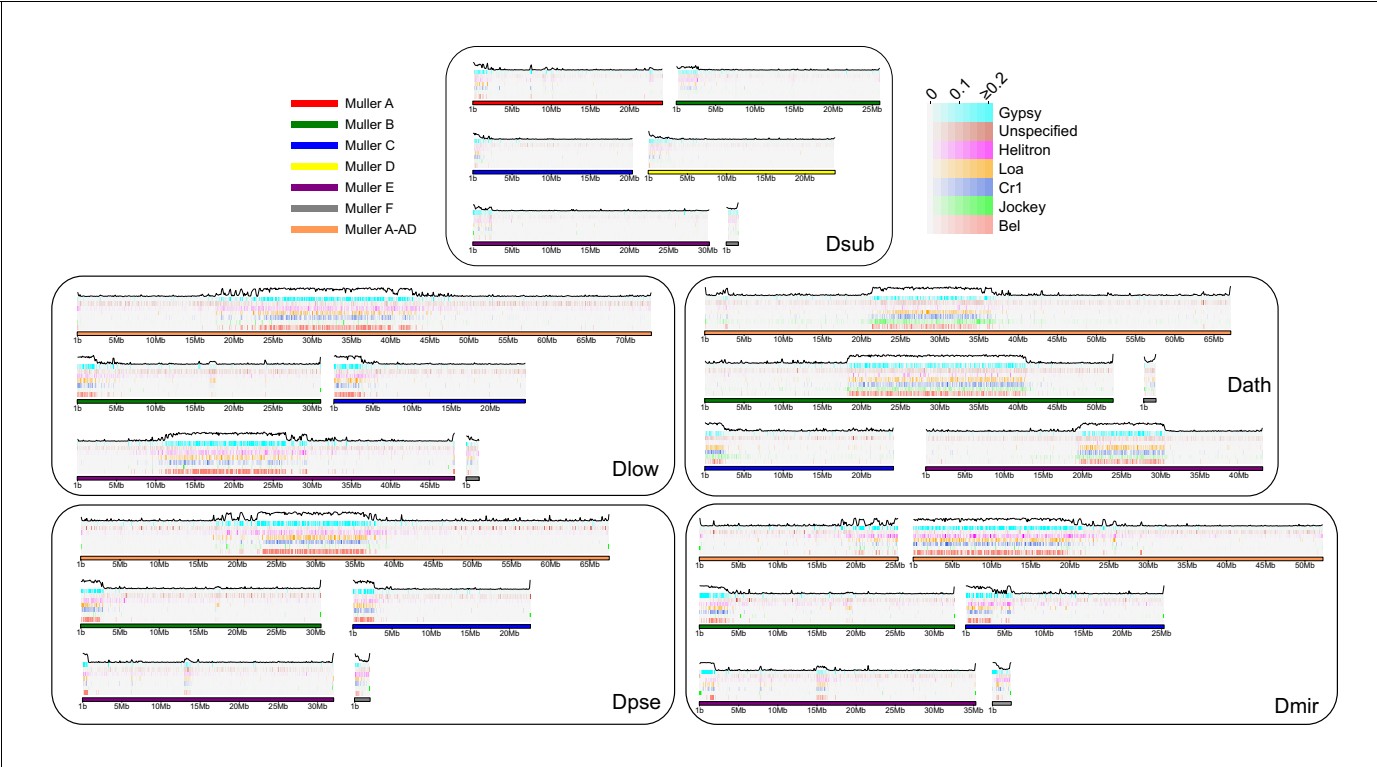

**Figure 8.** Transposable element evolution across the genome. Shown is the fraction of bases masked in 100 kb genomic windows for different transposable element families with the total TE fraction plotted above each chromosome.

DOI: https://doi.org/10.7554/eLife.49002.024

The following figure supplements are available for figure 8:

**Figure supplement 1.** Genomic distribution of transposable elements by species and Muller element.

DOI: https://doi.org/10.7554/eLife.49002.025

**Figure supplement 2.** De-novo estimates (dnaPipeTE) of transposable element frequencies in *D. subobscura*, *D. athabasca*, *D. lowei* and *D. pseudoobscura*.

DOI: https://doi.org/10.7554/eLife.49002.026

and a helitron element (HelitronN-1_DPe) are the most common repeats, with both repeats accounting for a similar fraction of repeat-masked bases in each genome (6–9% each; see *Table 3*).

As expected, the vast majority of repeats accumulate within pericentromeric regions (*Figure 8*), but some elements are predominantly found along chromosome arms (*Figure 8—figure supplement 1*). For example, *Gypsy8* appears enriched in euchromatic arms and several unspecified TEs are only found scattered across euchromatic regions of each chromosome (*Figure 8—figure supplement 1*). We typically find that the same TE is the dominant transposable element for all pericentromeres within a species, even if they were formed at different time points. Thus, there is more similarity among pericentromeric TE families between chromosomes within a species than there is between homologous chromosomes between species (*Figure 8—figure supplement 1*). Overall, genomic patterns of TE abundance reflect the dynamic evolution of repeats across the genome and the quick turnover that can occur between species.

## Discussion

The centromere is a complex cellular structure responsible for proper chromosome segregation during cell division. In almost all eukaryotes, centromeres are found at specific locations along chromosomes and are composed of (or embedded in) blocks of satellite DNA that occasionally are very large (mega-bases in size). Yet, in spite of the fundamental importance of centromere function and the high conservation of most centromeric proteins (*van Hooff et al., 2017*), both the centromere-

specific histone cenH3, and the centromeric satellite DNA sequences can differ substantially even among closely related species (*Henikoff et al., 2001*; *Melters et al., 2013*; *Steiner and Henikoff, 2015*). Further, centromere size and position can change rapidly along the chromosome.

Classical and molecular cytogenetic studies have provided rich data on the karyotypes of thousands of species in the last century (including information on diploid numbers, and relative length and morphology of chromosomes), and have revealed a dynamic evolutionary history of chromosomes across eukaryotes. The different positions of centromeres along a chromosome were historically almost always interpreted as the result of pericentric inversions or complex rearrangements. However, comparative cytogenetic studies of chromosome synteny among species, within a phylogenetic context, have revealed that centromere location on homologous chromosomes may change with no concomitant change in DNA marker order. 'Centromere repositioning', that is, the movement of the centromere along the chromosome without marker order variation, was first given as a possible mechanism for karyotype evolution in a broad investigation of chromosomal evolution in 60 species of primates (*Dutrillaux, 1979*). Over the past few decades, numerous studies have found evolutionary new centromeres in different lineages, including fungi, insects, birds, and mammals (*O'Neill et al., 2004*; *Marshall et al., 2008*; *Rocchi et al., 2012*; *Scott and Sullivan, 2014*; *Schneider et al., 2016*; *Burrack et al., 2016*; *Tolomeo et al., 2017*). Some of the observed karyotype reshuffling may be due to the inheritance of neo-centromeres (*Amor et al., 2004*) while some may be the product of successive pericentric inversions (*Rocchi et al., 2012*). However, almost all studies have used cytological techniques with limited resolution to track karyotype evolution at the DNA sequence level.

Drosophila, the workhorse of classical and molecular genetics, has a long history and rich literature of karyotype evolution (*Muller, 1940*; *Patterson and Stone, 1952*). Inversions are ubiquitous in flies. Thus, while dozens of transitions between telocentric and metacentric chromosomes have been documented in Drosophila (*Muller, 1940*; *Patterson and Stone, 1952*), pericentric inversions are typically invoked to explain the movement of centromeres between species (*Segarra et al., 1995*; *Schaeffer et al., 2008*). Here, we reconstruct karyotype evolution in flies using modern genomic techniques. Contrary to the common belief that pericentric inversions are the main mechanism for reshuffling karyotypes among different Drosophila lineages, our analysis suggests that centromere repositioning appears to play an important role in chromosome evolution in Drosophila. Note however, that we cannot refute complex structural rearrangements having contributed to the movement of centromeres across species.

## Neo-centromeres and de novo formation of centromeres

The location of the centromere on a given chromosome is inherited, such that syntenic centromeric loci are found in related species. Yet, centromeres are thought to be specified epigenetically, and can form at many positions along a chromosome (*Karpen and Allshire, 1997*; *Allshire and Karpen, 2008*). Neo-centromeres occur when kinetochores assemble de novo, at DNA loci not previously associated with kinetochore proteins. They are sometimes found at acentric chromosome fragments that arise by chromosome rearrangements, deletions, or amplifications and can restore the ability of an acentric chromosome fragment to segregate efficiently (*Burrack et al., 2016*). In rare cases, neo-centromeres form in otherwise normal chromosomes, without physical deletion of the native centromere, presumably following inactivation of the native centromere through unknown mechanisms (*Amor et al., 2004*; *Liehr et al., 2010*). More than 100 human neo-centromere locations have been identified (*Marshall et al., 2008*), and neo-centromere formation has been studied in several model systems including Drosophila (*Maggert and Karpen, 2001*), fungi (*Ishii et al., 2008*; *Burrack et al., 2016*) or chicken cell lines (*Shang et al., 2013*). Detailed analysis of neo-centromeres in humans and other species has shown that their formation is a rare and usually deleterious event, yet neo-centromeres can occur at many different regions across the genome. Sites of neo-centromere formation do not share a unique DNA sequence but may have in common other features that predispose to neo-centromere formation, such as increased AT content (*Amor and Choo, 2002*; *Amor et al., 2004*; *Marshall et al., 2008*). Fixations of neo-centromeres within a species result in the formation of 'evolutionary new centromeres'. Evolutionary new centromeres can be relatively frequent in some groups. In macaque, for instance, nine out of 20 autosomal centromeres are evolutionarily new; in donkey at least five such neo-centromeres originated after divergence from the zebra, in less than 1 million years. The establishment of new centromeres is often associated with an accompanying

expansion of satellites at the new centromere location and loss of large satellite arrays at the former location (*Ventura et al., 2007*). Although most human neo-centromeres are pathological chromosome aberrations that are evolutionarily rare and transient, it has become apparent that on occasion, the shift of a centromere to a new position becomes fixed, resulting in a permanent karyotypic change (*Cardone et al., 2007*).

Thus, detailed functional studies on neo-centromeres combined with comparative cytogenetic investigations of evolutionary new centromeres support that centromere repositioning may be important for karyotype evolution. As noted, however, investigations of these phenomena at the DNA level across species are lacking. Despite dramatic advances in sequencing technology, the repeat structure of centromeres has banished them to the last frontier of the genome to be deciphered for most organisms. Our study is the first to track the evolution of the centromere in flies at the sequence level, using high-quality genomic assemblies. Our comparative analysis allows us to reconstruct the entire life cycle of complex centromeres, including their formation, maturation and degradation.

## Seeding of new centromeres

We find that new centromeres in flies of the *obscura* group were formed in initially anonymous regions (*Figure 5*). The most striking feature of seed regions is their increased AT-content (*Figure 5*, *Figure 5—figure supplement 2*), which resembles observations from evolutionary novel centromeres in primates (*Federico et al., 2017*), or neo-centromeres in humans (*Amor and Choo, 2002*; *Amor et al., 2004*; *Marshall et al., 2008*) or chicken (*Shang et al., 2013*). Such (neo)centromeres typically arise in euchromatic regions that are otherwise unremarkable with respect to sequence composition but are reported to predominantly emerge in ancestral regions that are gene-poor and AT-rich. The lack of any obvious sequence features involved in the formation of new centromeres confirms the notion that the origin of a neo-centromere appears to be associated with epigenetic phenomena that are not yet well understood, and increased AT content may provide a more favorable disposition for the formation of novel centromeres. In other respects, these seed regions show no particular differences from bulk DNA sequences. In contrast to (neo)centromere formation in humans or other model systems, new centromeres in flies are seeded in ancestral regions that show typical gene density. Thus, centromere function in flies is not incompatible with gene activity, and we find that genes embedded in pericentromeric heterochromatin show normal levels of transcriptional output. This is surprising, since flanking regions of pericentromeric genes show strong enrichment for silencing chromatin marks, yet these epigenetic silencing marks do not spread over promoter regions or gene bodies in neo-centromeres in *D. athabasca*.

## Maturation of centromeres

Mature eukaryotic centromeres are composed of arrays of satellite DNA frequently surrounded by clusters of transposable element islands (*She et al., 2004*), or the other way round (*Talbert et al., 2018*; *Chang et al., 2019*), and are embedded in silencing heterochromatin. The vast majority of evolutionary new centromeres that have been investigated in primates possess a heterochromatic block similar to normal centromeres. FISH studies in macaque, for instance, have shown that all the nine novel centromeres formed in this lineage have large blocks of alpha-satellite DNA indistinguishable from other macaque centromeres (*Ventura et al., 2007*). Detailed analysis of one of these new centromeres has shown that it was seeded in a gene dessert, suggesting that the absence of genes may have been a prerequisite for survival of this centromere (*Ventura et al., 2007*). However, the molecular nature and sequence composition of new centromeres is generally unknown.

Our study is the first that bridges this gap, and we can directly infer the molecular and epigenetic changes accompanying the maturation of a centromere. In contrast to findings in macaques, the newly formed centromeres in *Drosophila* species emerged in gene-rich sequences. We further demonstrate that their establishment was accompanied by the acquisition of species-specific arrays of centromere-associated satellite DNA as well as clusters of pericentromeric transposable elements. In particular, the seeding domains for new centromeres in *D. subobscura* show a slight increase in both gene and repeat density but are devoid of arrays of satellite DNA. Following the seeding of a novel centromere on Muller A, B and E, these regions expanded dramatically in size. Most of this size increase appears to be driven by an accumulation of transposable elements. Different TEs

accumulate at pericentromeres in different fly species, but we typically find the same element as the dominant TE for all centromeres within a species. Thus, genomic patterns of TE abundance reflect the dynamic evolution of repeats.

Interestingly, we find taxon-specific satellites in each group that may represent the centromeric repeat. In our assemblies, we recovered a shared putative satellite repeat in *D. miranda*, *D. pseudoobscura* and *D. lowei* (a 21mer) and a 99mer unique to *D. miranda*, and a different 160 bp satellite in *D. athabasca*. Each of these satellites was enriched in the (peri)centric region of most chromosomes in our assemblies, suggesting that our genome assemblies may cover at least part of the centromeric satellite, and localization of these satellites near the centromere was confirmed by FISH. Only in *D. subobscura*, the species with the ancestral karyotype and putatively older (peri)centromeres, we failed to assemble a highly abundant repeat motif as a candidate for the centromeric satellite. Inspection of raw reads identified a highly abundant 12mer that was collapsed in our assembly but localized to the centromere in this species. Thus, long-read sequencing data allowed us to assembly (parts of the) younger, more heterogeneous pericentromeres, but not the older pericentromere of *D. subobscura*, which may be more homogenous (i.e. the satellite repeats may be interrupted by fewer TEs; see below). Note, however, that additional experiments are necessary to confirm the functional centromere in each species (*Talbert et al., 2018*; *Chang et al., 2019*).

Our data are thus consistent with a model where novel centromeres were formed in regions devoid of repeats, which began to accumulate TEs after their formation. TEs might provide the substrate for satellite DNA formation and trigger the initiation of species-specific satellites. Younger centromeres may be more heterogeneous and be a conglomerate of different TEs and satellites, and occasionally be interrupted by single-copy genes (and may resemble the younger pericentromeres observed in the *affinis* and *pseudoobscura* group). Homogenizing forces, such as gene conversion, could then operate to erode heterogeneity and complexity of satellites, leading to the evolution of simple and highly homogeneous satellite arrays observed in many old centromeres (as possibly found in *D. subobscura*).

Epigenetic models of centromere specification stress the importance of cenH3 nucleation in formation of a centromere (*Karpen and Allshire, 1997*; *Allshire and Karpen, 2008*). We show that the region that became part of the mature pericentromere is not limited to a single nucleation site (that is, encompassing only a few nucleosomes). Instead, the seed region spreads over several 100 kb for each of the newly formed pericentromeres, containing dozens of genes each. In the mature pericentromere, these regions are now many Mb in size due to invasion of TEs, and the genes are now embedded in large stretches of repeats. Silencing heterochromatin is found around many ancestral repeat-rich centromeres, and genes mapping to seed regions are at risk of silencing by successive heterochromatization. Indeed, we find that silencing heterochromatin has spread across the repetitive new pericentromeres and formerly euchromatic genes are now embedded in pericentromeric heterochromatin. However, silencing marks are absent at promoter regions, and do not seem to spread across gene bodies despite drastic enrichment at intergenic regions directly flanking these pericentromeric genes. In line with a lack of repressive chromatin marks, we find no reduction in expression levels of genes within pericentromeric regions. How spreading of heterochromatin across gene bodies is prevented is not clear and will require future detailed experimental investigation.

## Centromere deactivation

The kinetochore is formed on the centromere, and the presence of multiple centromeres on a single chromosome can cause abnormal spindle-chromosome interaction resulting in chromosome missegregation or chromosome breakage during cell division (*Brock and Bloom, 1994*). Having one and only one centromere per chromosome is thus essential for proper chromosome segregation. Centromere repositioning creates a novel centromere, suggesting that a single chromosome contains multiple latent centromeres. Chromosomes with multiple centromeres, such as dicentric chromosomes, can be stabilized by centromere inactivation, a process which reestablishes monocentric chromosomes (*Sato et al., 2012*). Studies of artificial dicentric centromeres in fission yeast demonstrated that the presence of a dicentric chromosome causes cell-cycle arrest in interphase. Epigenetic centromere inactivation could stabilize dicentric chromosomes, and consequent heterochromatinization from pericentric repeats to the central domain could prevent reactivation of the inactivated centromere (*Sato et al., 2012*). However, little is known about this process in naturally occurring dicentric chromosomes (*Cech and Peichel, 2016*).

Our high-quality genome sequences allow us to recover the remnants of former centromeres in flies in the *pseudoobscura* group, and we can reconstruct the stepwise disassembling of a repeat-rich centromere. In flies of the *pseudoobscura* group, both Muller B and E reverted to telocentric chromosomes, and phylogenetic dating suggests that this transition occurred 9–14 MY ago for Muller B, and 3–9 MY for Muller E. A pericentric inversion appears to have moved a small part of the pericentromere on Muller B, and a new centromere was probably created on Muller E, leaving behind most or all of the metacentric centromere (*Figure 5*). Our sequence data demonstrate how redundant centromeres can become inactivated and lost rapidly. We detect major re-arrangements at these former centromeres, and both have shrunken dramatically in size, through purging of repetitive DNA. The repeat content of both paleocentromeres is elevated only slightly compared to background regions. Interestingly, however, we find that both paleocentromeres are still strongly enriched for heterochromatin (*Figure 5*). In fission yeast, inactive centromeres are prevented from reactivation by heterochromatinization (*Sato et al., 2012*), and it is possible that heterochromatin formation at paleocentromeres in *pseudoobscura* flies is necessary to inhibit the assembly of a functional centromere. This is accompanied by removal of centromere-specific sequences, and we detect both rapid turnover of centromere-associated satellites between species, and no satellite enrichment at the decaying paleocentromeres on Muller B and E.

Dicentric chromosomes can result from centromere repositioning events, or chromosome fusions. As demonstrated by the extensive variation in chromosome number among species, chromosome fusion (and fission) events are common during evolution. For instance, in multicellular eukaryotes, chromosome number ranges from a single pair of chromosomes per cell in the Jack jumper ant (*Crosland and Crozier, 1986*) to 630 pairs of chromosomes in the Adders tongue fern (*Khandelwal, 1990*). Chromosome numbers can also vary substantially among closely related species (*The Tree of Sex Consortium, 2014*). The diversity of karyotypes across eukaryotes suggests that dicentric chromosomes accompanied by inactivation of one centromere may be common and may leave distinct footprints in the genomic make-up of a species. In our initial analysis of the *D. miranda* genome, we noticed the occurrence of repeat islands in this species, but their significance was unclear (*Mahajan et al., 2018*). Our comparative analysis now clearly demonstrates that this unusual sequence feature was in fact the legacy of an inactivated centromere, and repeat-islands in other genomes may likewise present remnants of former centromeres. Our study thus highlights that a full understanding of genome structure and function is only possibly in an evolutionary framework.

## What drives centromere relocation?

Reasons for centromere relocation could be manifold. The function of the ancestral centromere could become compromised by mutations, such as a loss of centromeric satellites, or insertions of transposable elements in the centromere (*Roizès, 2006*). This could result in the activation of a latent novel centromere, and inactivation of the ancestral one. Support for this model comes from investigations of neo-centromeres in humans, which were found on a chromosome with a substantially reduced size of the alpha satellite array on its ancestral chromosome compared to its wildtype size (*Hasson et al., 2011*). Novel centromeres could also become fixed by meiotic drive. Female meiosis is inherently asymmetric: only one of four meiotic products is transmitted to the next generation via the oocyte. Selfish genetic elements can exploit this asymmetry to preferentially transmit themselves, and a novel centromere that outcompetes the ancestral one may invade a population (*Henikoff et al., 2001*). In maize, non-centromeric blocks of satellite-DNA, or ''knobs,'' can transmit themselves more frequently than expected through asymmetric female meiosis, but not symmetric male meiosis. Knob-containing Abnormal 10 chromosomes can outcompete Normal 10 chromosomes and sponsor drive of other knob-containing chromosomes that are incapable of driving on their own (*Rhoades and Dempsey, 1966*). Invasion of such a drive system could lead to the simultaneous replacement of centromeres (or centromere satellites) on multiple chromosomes. Finally, de novo centromere formation may be linked to chromosome breakage. CENP-A (the centromere-specific histone H3 variant) was shown to be recruited to DNA double-strand breaks (*Zeitlin et al., 2009*) and a neo-centromere could emerge because of the presence of CENP-A at the breakpoint. We find that new centromeres are predominantly seeded in AT-rich regions, and several studies have suggested that AT-rich sequences are prone to breakage (*Zhang and Freudenreich, 2007*; *Kato et al., 2012*). This could provide a link between neo-centromere seeding and AT content through elevated rates of chromosome breakage.

## Evolution of centromere-associated satellites

Rapid evolution of centromeric satellite sequences has been observed across metazoan lineages. This rapid evolution is attributed to processes such as molecular drive, leading to the homogenization and fixation of a variant (or subset of variants) across a repeat array (*Dover, 1982*; *Dover et al., 1982*), and both genetic conflict and centromere drive (*Henikoff et al., 2001*) can lead to rapid diversification of repeat families between species. Our high-quality genome assemblies combined with bioinformatics approaches to detect satellite DNA allowed us to identify centromere-associated repeats in several species. Intriguingly, we find that the major satellite repeat can change rapidly between closely related species. For example, we find a 99 bp satellite in *D. miranda* that is (mostly) absent in *D. pseudoobscura* or *D. lowei*, suggesting that this sequence invaded the *D. miranda* genome in the last 1–2 MY ago and became the dominant centromere-associated satellite. *D. guanche* has a 290 bp satellite at its centromere, which is rare and not associated with the centromere in *D. subobscura* (*Bachmann et al., 2009*; *Puerma et al., 2018*; *Karageorgiou et al., 2019*). Thus, new satellites can form rapidly de novo and quickly become the dominant repeat within a species.

How do species-specific satellites emerge? Models of satellite evolution suggest two stages in the emergence of new satellites: Amplification processes generate small tandem sequences, and some of these sequences expand to large arrays by unequal exchange (*Stephan and Cho, 1994*). Several studies have found that TEs may contribute to the formation of novel satellite arrays. A stunning example of the birth of a satellite from a TE has been shown in *D. virilis* (*Dias et al., 2014*), where a foldback transposon provided internal tandem repeats that acted as 'seeds' for the generation and amplification of satellite DNA arrays. Furthermore, TEs from *D. melanogaster* were found to commonly form tandem dimers, which have the potential to provide source material for future expansion into satellite arrays (*McGurk and Barbash, 2018*). Thus, the genesis of satellites from TE insertions offers a possible explanation for the origination of species-specific centromere satellites. Transposons may start to accumulate on a new centromere and provide a substrate for the formation of satellite arrays either through their internal repeats structure, or by double insertion of a transposon into the same site, creating tandem TEs. Such tandem dimers can then expand into large arrays by recombination events. Indeed, many copies of the 99 bp satellite repeat identified in *D. miranda* show similarity to transposable elements in the *Repbase* repeat library, and future detailed study will help to decipher the evolutionary history of species-specific satellites in this species group.

Our data demonstrate that repetitive sequences can turn over rapidly between species and trigger major genomic responses. Recent data from hybrid model systems have shown that incompatibilities between genomes can be complex, and include interactions among heterochromatin, repeats or repetitive centromeres. It will be of great interest to identify the role rapidly evolving centromeres play in the establishment of genomic incompatibilities among closely related taxa and in the formation of new species (*Brown and O'Neill, 2010*).

# Materials and methods

**Key resources table**

| Reagent type (species) or resource | Designation | Source or reference | Identifiers | Additional information |
|---|---|---|---|---|
| Strain, strain background (*Drosophila subobscura* male and female) | 14011–0131.10 | National Drosophila Species Stock Center (Cornell University) | stock center number: 14011–0131.10 | |
| Strain, strain background (*Drosophila pseudoobscura* male and female) | MV2-25 | National Drosophila Species Stock Center (Cornell University) | stock center number: 14011–0121.94 | |
| Biological sample (*Drosophila lowei*) | Jillo6 | isofemale line (deceased) | | |

*Continued on next page*

*Continued*

| Reagent type (species) or resource | Designation | Source or reference | Identifiers | Additional information |
|---|---|---|---|---|
| Biological sample (*Drosophila athabasca* EA) | PA60 | isofemale line (deceased) | | |
| Biological sample (*Drosophila athabasca* EB) | NJ28 | isofemale line (deceased) | | |
| Commercial assay or kit | TruSeq Stranded RNA kit | Illumina | cat # 20020595 | |
| Commercial assay or kit | TruSeq DNA Nano Prep kit | Illumina | cat # 20015965 | |
| Commercial assay or kit | DNeasy Kit | Qiagen | cat # 69504 | |
| Commercial assay or kit | Blood and Cell Culture DNA Midi Kit | Qiagen | cat # 13343 | |
| Commercial assay or kit | Gentra Puregene Tissue Kit | Qiagen | cat # 158667 | |
| Commercial assay or kit | Ligation sequencing kit | Nanopore | SQK-LSK108 | |
| Commercial assay or kit | Rapid sequencing kit | Nanopore | SQK-RAD004 | |
| Commercial assay or kit | Quick DNA plus Midi kit | Zymo | cat # D4075 | |
| Commercial assay or kit | Ligation sequencing kit | Nanopore | SQK-LSK109 | |
| Commercial assay or kit | SMARTer Universal Low Input DNA-seq kit | Takara/Rubicon Bio | R400676 | |
| Chemical compound, drug | H3K9me3 antibody | Diagenode | | |
| Software, algorithm | canu | *Koren et al., 2017* | | |
| Software, algorithm | MUMmer | *Kurtz et al., 2004* | | |
| Software, algorithm | WTDBG2 | *Ruan and Li, 2019* | | |
| Software, algorithm | minimap2 | *Li, 2018* | | |
| Software, algorithm | Bandage | *Wick et al., 2015* | | |
| Software, algorithm | BWA | *Li and Durbin, 2009* | | |
| Software, algorithm | SAMtools | *Li et al., 2009* | | |
| Software, algorithm | bedtools | *Quinlan and Hall, 2010* | | |
| Software, algorithm | QUIVER | *Chin et al., 2013* | | |
| Software, algorithm | PILON | *Walker et al., 2014* | | |
| Software, algorithm | RACON | *Vaser et al., 2017* | | |
| Software, algorithm | Juicebox | *Durand et al., 2016a* | | |
| Software, algorithm | Juicer | *Durand et al., 2016b* | | |
| Software, algorithm | 3D-DNA | *Dudchenko et al., 2017* | | |
| Software, algorithm | GATK Unified Genotyper | *DePristo et al., 2011* | | |
| Software, algorithm | REPdenovo | *Chu et al., 2016* | | |
| Software, algorithm | RepeatMasker | *Smith et al., 2005* | | |
| Software, algorithm | MAKER | *Campbell et al., 2014* | | |

*Continued*

| Reagent type (species) or resource | Designation | Source or reference | Identifiers | Additional information |
|---|---|---|---|---|
| Software, algorithm | HiSat2 | *Kim et al., 2015* | | |
| Software, algorithm | StringTie | *Pertea et al., 2015* | | |
| Software, algorithm | BUSCO | *Simão et al., 2015* | | |
| Software, algorithm | Tandem Repeat Finder | *Benson, 1999* | | |
| Software, algorithm | k-Seek | *Wei et al., 2014*; https://github.com/weikevinhc/k-seek | | |
| Software, algorithm | TideHunter | *Gao et al., 2019* | | |

## Long read sequencing and contig assembly

For each Drosophila genome, we followed a similar assembly approach to *Mahajan et al. (2018)*. Briefly, long-read sequencing data were used to generate sequence contigs, Illumina data were used to polish the contigs and identify non-target contigs (contamination), and Hi-C data were used to place contigs into chromosomal scaffolds. However, given differences in the long-read datatype (PacBio or Nanopore), the amount of polymorphism in the strains sequenced, the sex of the individuals used for sequencing, and differences in the genome architecture (such as repeat content) across species, slightly different species-specific adjustments were done during each assembly, to maximize the quality of each Drosophila genome. Below we give a brief summary of the assembly approach taken for each species. A detailed description of species-specific adjustments for each assembly and validation of our assemblies are given in Appendix 1.

### Drosophila athabasca

We extracted high molecular weight DNA from ~100 females of semispecies EA (isofemale line PA60) and EB (isofemale line NJ28) (*Wong Miller et al., 2017*) using a QIAGEN Gentra Puregene Tissue Kit. DNA fragments >100 kb (estimated from pulsed-field gel electrophoresis) were then sequenced using the PacBio RS II platform with 13 SMRT cells for each semispecies. This resulted in a total of 622,136 reads with an N50 read length of 17,517 bp for EB and 1,195,701 reads with an N50 read length of 12,314 for EA. PacBio reads were assembled using canu (version 1.6) (*Koren et al., 2017*) with a correctedErrorRate of 0.075 for both the EA and EB assembly. During contig assembly with canu (here and below) we explored various correctedErrorRates spanning recommendations to maximize contig length and minimize misassembly.

### Drosophila pseudoobscura

We sequenced males from the reference genome strain (Drosophila Species Stock Number: 14011–0121.94; MV2-25) using five Nanopore MinION flowcells (versions 9.4 and 9.5) with the ligation sequencing kit (SQK-LSK108) and the rapid sequencing kit (SQK-RAD004). We combined the total output from these five runs, 1,250,165 filtered reads with an N50 read length of 18,209 bp with published female derived Nanopore reads of the same line (*Miller et al., 2018*). We were particularly interested in assembling Y chromosome contigs and thus took an approach to enrich our assembly for heterochromatic regions of the genome similar to *Chang and Larracuente (2019)*. We sequentially assembled the autosomes, the X (Muller A-AD), and the Y chromosome, each time subtracting out reads that mapped uniquely to that specific genomic partition. Remaining reads were used in subsequent assemblies. Nanopore reads were mapped to assemblies using minimap2 (*Li, 2018*) and autosomal and X-linked contigs were identified via comparison with *D. miranda* (*Mahajan et al., 2018*) using MUMmer (*Kurtz et al., 2004*). To first assemble the autosomes, all reads >10 kb were selected using FiltLong (https://github.com/rrwick/Filtlong) and assembled using canu (version 1.7), with the -fast option and a correctedErrorRate of 0.144. The X-linked contigs were assembled using identical settings. The Y-linked assembly was performed using only male derived reads and a correctedErrorRate of 0.166. To distinguish between putative Y-linked contigs and unplaced autosomal

contigs in the final assembly, contigs with >1.5 male/female mean Illumina coverage (described below) were considered putatively Y-linked, with the remaining left unplaced.

## Drosophila subobscura

We obtained the isofemale line 14011–0131.10 from the National Drosophila Species Stock Center (Cornell University). We extracted high molecular weight DNA form ~100 females using a Zymo quick DNA plus Midi kit (cat # D4075) following the manufacturers recommendations. Isolated DNA was then size selected for fragments >15 kb using BluePippin (Sage Science) and was sequenced on the Nanopore MinIOn using the SQK LSK-109 ligation protocol and two flow cells (version R 9.4.1). This resulted in a total of 1,689,885 filtered reads with an N50 read length for the combined reads of 17,390 bp. Initial assemblies of our data following methods outlined above resulted in many large misassembled contigs likely due to an observed enrichment of chimeric Nanopore reads in our data. To deal with this issue we assembled reads using canu (version 1.8) and modified several parameters to increase stringency (correctedErrorRate = 0.065 corMinCoverage = 8 batOptions='-dg 3 db 3 -dr 1 -ca 500 -cp 50' trimReadsCoverage = 4 trimReadsOverlap = 500 genomeSize = 150 m). The resulting error-corrected reads were then assembled into contigs using the WTDBG2 assembler (*Ruan and Li, 2019*) with default settings.

## Drosophila lowei

Flies were collected from Woodland Park, CO in August of 2018 (elev. 8500 ft.) to establish multiple isofemale lines. Due to difficulties in maintaining this species that would allow us to inbreed and reduce heterozygosity prior to sequencing, we extracted DNA from a pool of ~60 F1 males from the most productive isofemale line, Jillo6. We extracted high molecular weight DNA using a Qiagen Blood and Cell Culture DNA Midi Kit (cat # 13343) and manufactures recommendations. The isolated DNA was size selected as above for fragments > 30 kb and was sequenced as above using the Nanopore MinIOn SQK LSK-109 protocol, on one R 9.4.1 flow cell. One sequencing run produced a total of 437,080 filtered reads with an N50 read length of 36,073 bp. To assemble the reads into contigs, we followed the methods outlined above for *D. subobscura*.

## Short read Illumina data

To aid in genome assembly steps outlined below, we generated Illumina short-read male and female whole genome sequence data for each species. We extracted DNA from a single male and female using a Qiagen DNeasy kit following manufacturers recommendations. DNA libraries were then prepared using the Illumina TruSeq Nano Prep kit and sequenced on a Hiseq 4000 with 100 bp PE reads. Plots of Illumina short read data were plotted using *circlize* (*Gu et al., 2014*).

## Identification of sex chromosome and non-target contigs

To identify sex chromosome contigs during assembly and flag putative non-target contaminant contigs (e.g., yeast, bacteria) for removal from our assemblies we interrogated each assembly graph output (.gfa file) from our canu assemblies using multiple metrics that were visualized in Bandage (*Wick et al., 2015*). These metrics included the mean, median and standard deviation of male and female Illumina coverage by contig, the ratio of female/male coverage, the median long-read coverage (PacBio or Nanopore) and the GC-content (%). We also compared the connectivity of contigs in the assembly graph since non-target contigs are typically isolated from the target species graph. We also identified the top two BLAST hits for all contigs <1 MB. To estimate Illumina coverage-based metrics (above), short and long reads were mapped to each assembly prior to visualization using BWA MEM (*Li and Durbin, 2009*). Downstream files were processed with SAMtools (*Li et al., 2009*) and coverage estimated using bedtools genomecov (*Quinlan and Hall, 2010*).

## Genome polishing

The *D. athabasaca* EA and EB PacBio-based assemblies were individually polished using the SMRT analysis software pipeline and two rounds of QUIVER (*Chin et al., 2013*) and two rounds of PILON (version 1.22) (*Walker et al., 2014*). EA and EB were polished with 47× and 48× coverage of 100 bp PE Illumina reads, respectively. The *D. pseudoobscura, D. lowei* and *D. subobscura* Nanopore-based assemblies were polished using three rounds of RACON (version v1.2.1) (*Vaser et al., 2017*)

followed by two rounds of PILON using 115×, 47× and 40× mean coverage of Illumina reads, respectively.

## Hi-C libraries and genome scaffolding

To scaffold together contigs from our de novo genome assemblies (above) we used chromatin conformation capture to generate Hi-C data. The Hi-C libraries for *D. athabasca* EA and *D. pseudoobscura* were prepared following methods in *Lieberman-Aiden et al. (2009)* as described in *Mahajan et al. (2018)* using the restriction enzyme HindIII. Tissues used for Hi-C were from female third instar larvae for *D. athabasca* and male third instar larvae for *D. pseudoobscura*. For *D. subobscura* and *D. lowei*, in situ DNase Hi-C was performed on adult females and males, respectively, as described in *Ramani et al. (2016)* with a few modifications. Briefly, tissue was ground with a micropestle in 1.5 ml microtube with 250 µl PBS buffer and we rinsed the pestle with 750 µl of PBS buffer. The resulting 1 ml homogenate was mixed gently and passed through 100 µm Nylon mesh (sterile cell strainer) to remove debris. The homogenate was centrifuged at 2000Xg for 5 min at 4°C and the cell pellet fixed with 1 ml of 4% formaldehyde at RT for 10 min, gently inverting tubes every 2 min. The resulting cells were treated as per *Ramani et al. (2016)* from steps 4 to 74. DNA Libraries were prepared with an Illumina TruSeq Nano library Prep Kit and sequenced on a Hiseq 4000 with 100 bp PE reads.

To identify Hi-C contacts for scaffolding our genome assemblies we used Juicer (*Durand et al., 2016b*) to map reads and generate Hi-C contact maps to orient contigs based on 3D interactions (*Supplementary file 10*). We then used the 3D-DNA pipeline (*Dudchenko et al., 2017*) to scaffold contigs and the resulting hic and assembly files were visualized using Juicebox (*Durand et al., 2016a*). To confirm scaffolding quality and to correct potential mistakes we first compared chromosome assignments from 3D-DNA-scaffolded chromosomes to other closely related species with chromosome-level assemblies using MUMmer (i.e., *D. miranda* and *D. guanche*; *Mahajan et al., 2018*; *Puerma et al., 2018*). Second, we compared female and male whole genome sequencing coverage across 3D-DNA-scaffolded chromosomes to confirm predicted patterns of coverage for sex-linked chromosomes and autosomes. Finalized assemblies were scaffolded together with 3D-DNA scripts and contigs joined with 500 N's (*Figure 2—figure supplement 3*).

## Hybrid *D. athabasca* genome assembly

Residual heterozygosity in our EA and EB isofemale lines initially led to fragmented assemblies of particular chromosomes. Therefore, we created a hybrid of the two *D. athabasca* assemblies that combined the most contiguous Muller elements from each into one final assembly. Initially, Muller C (which is fixed as a neo-sex chromosome in EB) was highly fragmented in our superior EB assembly and Muller E was highly fragmented in our less contiguous EA assembly. Therefore, we combined the Muller C assembly from EA with the rest of the genome for EB. To make Muller C more 'EB-like' we mapped our female Illumina data (above), called variants using GATK's UnifiedGenotyper (*DePristo et al., 2011*), and used FastaAlternateReferenceMaker to replace EA variants with EB variants. All three assemblies are made available.

## Identifying repeats and masking genomes

We used REPdenovo (*Chu et al., 2016*) to annotate repeats in each of our draft genome assemblies prior to gene annotation. We first ran REPdenovo on our raw Illumina sequencing reads derived from the same sex as the genome assembly using the following parameters, MIN_REPEAT_FREQ 4, RANGE_ASM_FREQ_DEC 2, RANGE_ASM_FREQ_GAP 0.8, K_MIN 30, K_MAX 50, K_INC 10, K_DFT 30, ASM_NODE_LENGTH_OFFSET −1, MIN_CONTIG_LENGTH 100, IS_DUPLICATE_REPEATS 0.85, COV_DIFF_CUTOFF 0.5, MIN_SUPPORT_PAIRS 20, MIN_FULLY_MAP_RATIO 0.2, TR_SIMILARITY 0.85, RM_CTN_CUTOFF 0.9, with READ_DEPTH and GENOME_LENGTH settings for each species specified based on preliminary genome assemblies and mapping based coverage estimates (above). We then used TBLASTN with the parameters -evalue 1e-6, -numalignments 1, and -numdescriptions 1 to BLAST translated *D. pseudoobscura* genes (r3.04) to our REPdenovo repeat library and eliminated any hits. For each species, we concatenated our REPdenovo repeats with the Repbase *Drosophila* repeat library (downloaded March 22, 2016, from www.girinst.org) and then used this concatenated fasta file to mask each genome with RepeatMasker version 4.0.7

(*Smith et al., 2005*) using the -no_is and -nolow flags. To characterize specific classes of repeats and their distibutions across the genome we used a curated repeat library for obscura group flies (*Hill and Betancourt, 2018*). Plots were generated using KaryoploteR (*Gel and Serra, 2017*).

## Genome annotation

To annotate our genome assemblies we used the MAKER annotation pipeline (*Campbell et al., 2014*). For *D. athabasca* and *D. pseudoobscura*, we used gene expression data (RNA-seq) to generate transcriptomes which were then used as ESTs during annotation. For *D. athabasca* we first extracted total RNA using the TRIZOL RNA isolation method and generated RNA-seq data using the Illumina TruSeq Stranded RNA kit. RNA was isolated for multiple tissues and developmental stages from both EA and EB. Individual RNA-seq libraries consisted of pools of tissue from multiple individuals and included ovaries, testes, male and female adult heads, male and female whole larvae, and male and female larval heads (*Supplementary file 1*). All RNA-seq libraries were sequenced using an Illumina HiSeq 4000 with 100 bp PE reads. Raw Illumina reads were aligned to the genome using HiSat2 with default settings and the -dta flag for transcriptome assembly (*Kim et al., 2015*). We then used StringTie (*Pertea et al., 2015*) to build semispecies specific transcriptome assemblies. Full-length transcripts were extracted from the transcriptome file using gffread and used downstream with MAKER. For *D. pseudoobscura* we used available gene expression data (RNA-seq) to create a de novo transcriptome as above. Gene expression data used to generate the transcriptome was from ovary, testis, male and female 3$^{rd}$ instar larvae, male and female whole adult, male and female carcass, male and female head, and female accessory glands (*Kaiser and Bachtrog, 2014*). To annotate our genome assemblies, we used published protein sets from *D. pseudoobscura* and *D. melanogaster* (FlyBase) in addition to our EST libraries. The *ab initio* gene predictors SNAP (*Korf, 2004*) and Augustus (*Stanke and Waack, 2003*) were used to guide the annotation and we retrained the gene predictors after the first run of MAKER. In total we ran two rounds of MAKER on *D. athabasca* and *D. pseudoobscura*. For *D. subobscura* and D. lowei, we lacked high-quality RNA-seq data to build de novo transcriptomes and therefore we annotated these genomes using just protein sets (above) and a single round of MAKER.

## Genome completeness and identification of Muller elements

To assess genome assembly completeness throughout the many assembly steps we used BUSCO (*Simão et al., 2015*) version 3.0.2 and the arthropod database (odb9). To identify the Muller elements in our assemblies and their conservation across Drosophila, we assigned 1055 complete and single copy BUSCOs in *D. melanogaster* (dmel r6.12) to their respective Muller element and then plotted their corresponding location in each genome assembly, allowing for duplication.

## Identification of obscura group orthologs

To identify orthologous genes and syntenic regions between the five genome assemblies we performed BLASTP reciprocal best hit searches between each species pair using Muller element specific proteins derived from our MAKER annotations (above). We used the blast_rbh.py script (*Cock et al., 2015*) and the gene coordinates were plotted using *genoPlotR* (*Guy et al., 2010*).

## Identification of putative centromeric satellite sequence

To identify tandem repeats and putative centromeric satellites we first used Tandem Repeat Finder (TRF) (*Benson, 1999*). We ran TRF on each genome assembly with the following settings, 2 7 7 80 10 50 500 f -d -m. The resulting output was parsed using the k_seek.pl perl script (*Wei et al., 2014*) to collapse identical repeats and create a multi-fasta of all unique kmers in the genome. To infer the centromeric satellite sequence we plotted the frequency of unique tandem repeat kmer lengths for each species. To identify locations of putative centromeric satellite sequences, we took all unique kmers from the inferred centromeric satellite kmer length for a given species (e.g., all related 99mers for *D. miranda*) and used Repeatmasker to mask the genome using default settings. To identify short satellite sequences we analyzed Illumina data using k-Seek (*Wei et al., 2014*) and the associated k_seek.pl script. *Drosophila guanche* Illumina data used in TRF and k-Seek analyses was downloaded from the NCBI (SRA accession, ERX2095402). To characterize tandem repeats directly from long reads we used TideHunter (*Gao et al., 2019*) with default settings and only kept

consensus repeats that were >4 bp. Since the TideHunter output includes a consensus repeat sequence for each read, similar repeats and HOR variants will be found repeatedly in the output. Therefore, we clustered all consensus repeats using CD-HIT (*Li and Godzik, 2006*) with a sequence identity cutoff of 90% and a 10 bp length cutoff to help condense the output. After clustering the consensus repeats, we then BLAST searched (default settings) the representative CD-HIT cluster sequences for putative centromeric satellite sequence identified from earlier TRF analyses This allowed us to identify centromeric satellite clusters and determined their relative contribution.

## Pericentromere H3K9me3 enrichment

To characterize H3K9me3 enrichment along each chromosome to help identify pericentromeric regions we generated ChIP-seq data for *D. athabasca*, *D. subobscura* and *D. lowei*, and analyzed published data for *D. pseudoobscura* (Gibilisco and Bachtrog *submitted*) and *D. miranda* (*Mahajan et al., 2018*). We generated ChIP-seq data from larval brains (*D. athabasca* EA and EB) or adult heads (*D. subobscura* and *D. lowei*) using the ultra-low-input micrococcal nuclease-based native ChIP and sequencing method (*Brind'Amour et al., 2015*) and an H3K9me3 antibody from Diagenode. *Drosophila melanogaster* was used as input for normalization. Libraries for sequencing were prepared using the SMARTer Universal Low Input DNA-seq kit (Takara/Rubicon Bio) and sequenced using 100 bp PE reads on a HiSeq 4000. To quantify enrichment, raw reads from each species were mapped to their respective genome using bwa mem, downstream files processed with SAMtools, and read counts were estimated in 50 kb windows using bedtools genomeCoverageBed. Counts were normalized using methods outlined in *Bonhoure et al. (2014)*. To create metagene plots we used metagene-maker (https://github.com/bdo311/metagene-maker) and normalized counts from 100 bp windows.

## Gene expression analyses

To characterize gene expression in *D. athabasca* we analyzed EB RNA-seq data from whole larva and pooled larval heads (above, *Supplementary file 1*). For *D. subobscura* we analyzed RNA-seq data generated from two pools of whole individuals (*Uppsalapool* and *Valenciapool*; *Porcelli et al., 2016*). For each species, all reads were aligned to their respective reference genomes using HiSat2 (*Kim et al., 2015*) and StringTie (*Pertea et al., 2015*) was used to estimate gene expression (FPKM). Downstream analyses were performed only on the subset of protein coding genes identified as orthologous between *D. subobscura* and *D. athabasca* (above).

## Fluorescent in situ hybridization

DNA oligos with 5′ modification or 5′ and 3′ modifications were used for in situ hybridization (*Supplementary file 6*). Tissue preparation and hybridization methods follow those of *Larracuente and Ferree (2015)*. Slides were imaged at the UC Berkeley Molecular Imaging Center using a Nikon Spinning Disk Confocal Microscope at 100X (with oil) with lasers 405, 488, 561, and 637 nm. Images were obtained using NIS-Elements software.

## Acknowledgement

We thank Kevin Wei for discussions on satellite evolution. Thanks to Alison Nguyen, Emily Chong and Lauren Gibilisco for help in generating data. We would also like to thank J J Emerson for technical assistance with PacBio data generation and Jill Bracewell for collecting *D. lowei* flies. This research was supported by NIH grants (R01GM076007, R01GM101255 and R01GM093182) to D Bachtrog and by NIH grant 5F32GM123764 to R Bracewell. This work used the Vincent J Coates Genomics Sequencing Laboratory at UC Berkeley, supported by NIH S10 OD018174 Instrumentation Grant.

## Additional information

### Funding

| Funder | Grant reference number | Author |
|---|---|---|
| National Institutes of Health | R01GM076007 | Doris Bachtrog |
| National Institutes of Health | R01 GM101255 | Doris Bachtrog |
| National Institutes of Health | R01GM093182 | Doris Bachtrog |
| National Institutes of Health | 5F32GM123764 | Ryan Bracewell |

The funders had no role in study design, data collection and interpretation, or the decision to submit the work for publication.

### Author contributions

Ryan Bracewell, Conceptualization, Data curation, Software, Formal analysis, Validation, Investigation, Visualization, Methodology, Writing—review and editing; Kamalakar Chatla, Matthew J Nalley, Data curation, Investigation, Methodology; Doris Bachtrog, Conceptualization, Resources, Data curation, Supervision, Funding acquisition, Validation, Investigation, Visualization, Writing—original draft, Project administration, Writing—review and editing

### Author ORCIDs

Doris Bachtrog  https://orcid.org/0000-0001-9724-9467

### Decision letter and Author response

Decision letter https://doi.org/10.7554/eLife.49002.054
Author response https://doi.org/10.7554/eLife.49002.055

## Additional files

### Supplementary files

- Supplementary file 1. DNA sequence data generated for this study.
DOI: https://doi.org/10.7554/eLife.49002.028

- Supplementary file 2. Drosophila strains used for genome assembly.
DOI: https://doi.org/10.7554/eLife.49002.029

- Supplementary file 3. BUSCO results for assembled genomes. For *D. athabasca*, BUSCO scores for the EB assembly are shown.
DOI: https://doi.org/10.7554/eLife.49002.030

- Supplementary file 4. Number of protein coding gene models from MAKER annotations for each genome assembly and Muller element.
DOI: https://doi.org/10.7554/eLife.49002.031

- Supplementary file 5. Average percentage of bases repeat-masked in each pericentromere.
DOI: https://doi.org/10.7554/eLife.49002.032

- Supplementary file 6. Inferred centromeric satellite sequence and fluorescent in situ hybridization probes.
DOI: https://doi.org/10.7554/eLife.49002.033

- Supplementary file 7. Comparison of gene density, repeat density, and GC-content (%), between seed and non-seed regions of Muller A, B and E in *D. subobscura. P*-values for comparisons, Mann-Whitney U.
DOI: https://doi.org/10.7554/eLife.49002.034

- Supplementary file 8. The most common repeat families in each species and amount of masked sequence (bp).
DOI: https://doi.org/10.7554/eLife.49002.035

• Supplementary file 9. The top 10 TE's from each Muller element and amount of masked sequence (bp).
DOI: https://doi.org/10.7554/eLife.49002.036

• Supplementary file 10. Hi-C data summary.
DOI: https://doi.org/10.7554/eLife.49002.037

• Transparent reporting form
DOI: https://doi.org/10.7554/eLife.49002.038

## Data availability

All sequencing data and assemblies have been deposited in GenBank (see Supplementary file 1 for all data and accession numbers).

The following dataset was generated:

| Author(s) | Year | Dataset title | Dataset URL | Database and Identifier |
|---|---|---|---|---|
| Bracewell R, Bachtrog D | 2019 | Genome sequencing and assembly of Drosophila | https://www.ncbi.nlm.nih.gov/bioproject/PRJNA545704 | NCBI BioProject, PRJNA545704 |

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

# Appendix 1

DOI: https://doi.org/10.7554/eLife.49002.039

For each Drosophila genome, we followed a similar assembly approach to *Mahajan et al. (2018)*. Briefly, long-read sequencing data were used to generate sequence contigs, Illumina data were used to polish the contigs and identify non-target contigs (contamination), and Hi-C data were used to place contigs into chromosomal scaffolds. However, given differences in the long-read datatype (PacBio or Nanopore), the amount of polymorphism present in the strains sequenced, the sex of the individuals used for sequencing, and differences in the genome architecture (such as repeat content) across species, slightly different species-specific adjustments were done during each assembly, to maximize the quality of each Drosophila genome. What follows is a description and characterization of steps taken during the assembly process for each species. In all cases, genome assemblies were initially interrogated using a variety of metrics to help determine the assembly quality and identify sex chromosome contigs and contaminant sequence (see Materials and methods). We used Bandage (*Wick et al., 2015*) to visualize our preliminary assemblies and metrics of interest were displayed overtop the assembly (*Appendix 1—figure 1*). By using Bandage, we were able to easily explore various assembly algorithm parameters and their impacts on the quality of the draft assembly. Further, we could filter the assemblies in an efficient manner and find additional contig associations in the graph that were further validated using Hi-C associations, and in some cases, contigs could be merged when appropriate (see below).

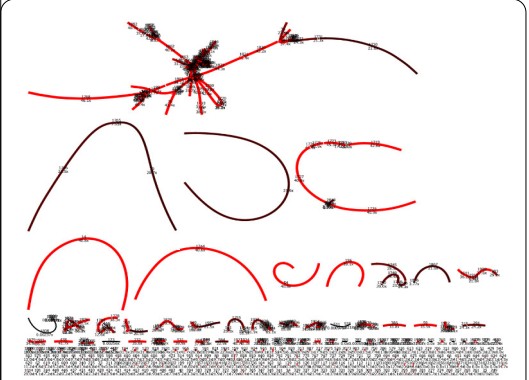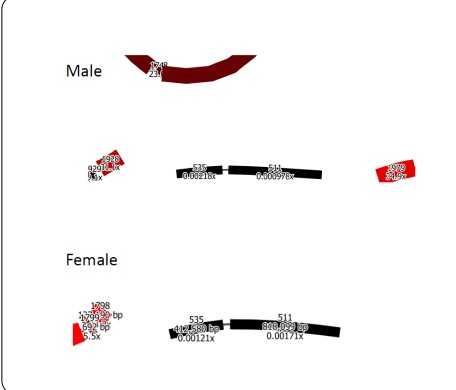

**Appendix 1—figure 1.** Bandage plot of a typical Drosophila genome assembly. The left panel is a visualization of the genome graph (.gfa file) from a canu assembly with the node name for each contig and Illumina coverage displayed in text overtop each contig. Each contig in the assembly is shaded by the amount of male Illumina whole genome sequencing coverage (see Materials and methods). In this example, red contigs are likely autosomal (~40×) while darker contigs have less coverage and indicate either putative sex chromosome contigs (~20×) or putative contaminant contigs (<<20×). (B) Shown is a zoomed in image of 2 nodes (535 and 511) in the assembly with exceptionally low male (top) and female (bottom) Illumina coverage (<0.1×). By also visualizing the top BLAST hits for these contigs (not shown), we were able to identify these contigs as belonging to an *Acetobacter* species and were thus contaminants marked for removal from the assembly. Contigs with exceptionally high Illumina coverage were also scrutinized thoroughly but these can arise for multiple reasons, including mtDNA contigs, collapsed regions of the target genome (e.g., rDNA genes or centromeric satellite sequence), or non-target contaminant contigs.

DOI: https://doi.org/10.7554/eLife.49002.040

## Drosophila miranda

The *D. miranda* genome was previously published, and the quality (both in terms of accuracy and contingency) of this assembly was confirmed with several orthogonal data sets including optical maps and extensive BAC clone sequencing (*Mahajan et al., 2018*). A detailed description of the assembly methods, including extensive QC at each assembly step are given in *Mahajan et al. (2018)*, and the assembly was also validated via statistical methods and short-read Illumina mapping (S3 Table in *Mahajan et al., 2018*). The quality and large-scale structural continuity of the *D. miranda* assembly was assessed by comparing it to sequenced BAC clones and optical mapping data. A total of 383 randomly selected BAC clones from a *D. miranda* male BAC clone library were shotgun sequenced (which should cover roughly 1/4 of the *D. miranda* genome), and 97% contiguously mapped to a unique position in the genome (see *Figure 2D*; S5 Table; S6 Table; S6 Fig in *Mahajan et al., 2018*). Similarly, most of the *D. miranda* genome was found to be covered by optical mapping data (S5 Fig and S4 Table in *Mahajan et al., 2018*). Thus, continuous and unique mapping of most BAC clones and coverage by optical reads confirmed the quality of the genome assembly.

Most importantly, BAC clone sequencing confirmed centromere assemblies in *D. miranda* (see S11 Fig from *Mahajan et al., 2018*), and BAC clone sequencing also confirm the assembly of 'paleocentromeres' in *D. miranda* (*Appendix 1—figure 2*). Additionally, the presence of heterochromatic repeat island on Muller B and Muller E in *D. miranda* (that is, the paleocentromeres) were confirmed by FISH in polytene chromosomes of *D. miranda* (see Figure 1C of *Mahajan et al., 2018*). Thus, our comprehensive and detailed validation in *D. miranda* demonstrated that our approach is accurate in assembling heterochromatic regions, and similar approaches were used to assemble the genomes of the other species that we report here.

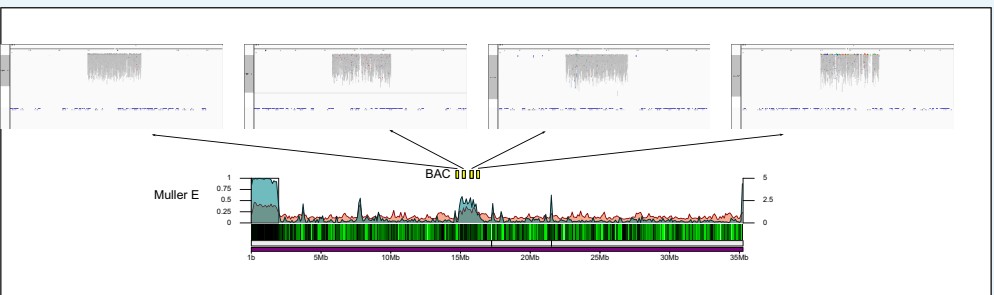

**Appendix 1—figure 2.** BAC clone sequencing confirms centromere and pealeocentromere assembly. Several independent BAC clones map to the assembly of our paleocentromeres in *D. miranda*.

DOI: https://doi.org/10.7554/eLife.49002.041

## Drosophila subobscura

Our *D. subobscura* assembly was our most contiguous assembly (*Appendix 1—table 1*) and had the fewest contigs in the initial canu/WTDBG2 assembly (*Appendix 1—table 1*). This is probably due to the low density of repetitive elements in this genome (see RESULTS), and others have noted that *D. subobscura* and close relative *D. guanche* were easier to assemble and produced more contiguous assemblies than other Drosophila species (*Puerma et al., 2018*; *Karageorgiou et al., 2019*). The subsequent polishing steps of the genome assembly with Racon and Pilon (*Walker et al., 2014*; *Vaser et al., 2017*) improved the assembly and at each round the number of single copy BUSCOs (*Simão et al., 2015*) increased, while the number of missing and fragmented BUSCOs decreased (*Appendix 1—table 1*). Hi-C scaffolding allowed us to orient contigs into chromosomes and all Muller elements were captured in <10 contigs, and typically just a few contigs made up the bulk of the chromosome (See RESULTS). Multiple small contigs showed aberrant Illumina coverage patterns and BLAST to *Acetobacter spp.* and *Saccharomyces cerevisiae* (used as Drosophila food and found in the gut). This contaminating sequence (eight contigs amounting to 2.2 Mb) was removed from the

assembly before final scaffolding (*Appendix 1—table 1*). In our assembly, the number of contigs per chromosome was slightly lower, and the total length of each chromosome similar to a recently published PacBio-based assembly of the *ch-cu* line (*Karageorgiou et al., 2019*). Importantly, the centromeric seed regions on Muller A, B and E were all assembled in contigs in our assembly. Interestingly, the seed region on Muller B (Chr U) is also associated with a well-known inversion breakpoint that results in a segregating inversion that plays a role in environmental adaptation (*Karageorgiou et al., 2019*). Our assembly captures this inverted chromosome and provides a resource for further study of this inversion in *D. subobscura*.

**Appendix 1—table 1.** Summary statistics and BUSCO results from the genome assembly process of *Drosophila subobscura*.

| | Canu/WTDBG2 only | Canu/WTDBG2 + Racon (3x) | Canu/WTDBG2 + Racon (3x) + Pilon (1x) | Canu/WTDBG2 + Racon (3x) + Pilon (2x) | Final Hi-C scaffolded Dsub_1.0 |
|---|---|---|---|---|---|
| N50 | 11,277,487 | 11,360,548 | 11,375,305 | 11,370,518 | * |
| Max Contig | 25,648,096 | 25,831,582 | 25,849,837 | 25,836,392 | * |
| Assembly Size | 128,396,075 | 129,296,034 | 129,434,338 | 129,376,892 | 126,232,139 |
| Number of Contigs | 62 | 61 | 61 | 61 | * |
| Complete BUSCOs | 987 | 1017 | 1057 | 1060 | 1062 |
| Complete Single Copy BUS-COs | 984 | 1011 | 1047 | 1050 | 1054 |
| Duplicated | 3 | 6 | 10 | 10 | 8 |
| Fragmented | 46 | 29 | 2 | - | - |
| Missing | 33 | 20 | 7 | 6 | 4 |
| % BUSCOs complete | 92.6% | 95.4% | 99.2% | 99.4% | 99.6% |

DOI: https://doi.org/10.7554/eLife.49002.042

## Drosophila athabasca

For *D. athabasca*, we produced assemblies for two different lines belonging to different semispecies (EA and EB). Flow cytometry estimates of genome size were ~210 Mb for both EA and EB.

### EB assembly

For the initial canu assembly we used the setting genomeSize = 210 m. Using Bandage and BLAST we found 10.4 Mb of non-target sequence (primarily *Acetobacter* and *Sacchoramyces*) and this sequence was removed. Nodes that showed unambiguous associations in the canu graph and had Hi-C associations during preliminary analyses were merged in the assembly graph. We merged the nodes 2416 and 2418 (total 34,554,387 bp), 2438 and 2440 (total 23,432,006 bp), 2396, 2474, 2476, and 2400 (total 22,894,561 bp), 2409 and 2411 (total 9,116,924 bp), 58881, 58882, and 2424 (total 1,243,185 bp). We also split one contig (node 2401), which was a chimeric assembly that showed Illumina coverage patterns suggesting it being both autosomal and X-linked and which showed split autosomal/X-linked associations in our preliminary Hi-C analyses. For Muller C in the EB assembly, residual heterozygosity led to a fragmented assembly with many regions where fragments of both haplotypes were assembled. To try to improve contiguity for this Muller element and collapse heterozygous regions, we took all reads that mapped uniquely to contigs identified as belonging to Muller C and reassembled them using Minimap2/Miniasm

(*Li, 2018*), which in our experience often collapses heterozygous regions. We used the miniasm flags -s 1000 -d 500000 and inspected the resulting graph file (.gfa) and merged contigs in the graph with unambiguous connections. We merged nodes utg000002l, utg000034l, utg000039l, and utg000029l (total 3,929,811 bp), utg000010l and utg000036l (total 6,966,474 bp) and these plus the remaining contigs were used in place of the canu-based contigs of Muller C for final Hi-C scaffolding. Hi-C scaffolding clustered 22 Muller C contigs, but because of the shorter contig lengths and ambiguities in their placement due to the Hi-C being derived from the other semispecies, we assigned each contig to Muller C and named them Muller_C 1–22.

Our EB assembly statistics (*Appendix 1—table 2*) improved at every step of the process and the number of complete BUSCOs increased with Quiver and Pilon genome polishing. Additional non-target sequence (total 3 Mb) was identified and removed after our Quiver runs (*Appendix 1—table 2*). The final scaffolded assembly contained long contigs that made up the bulk of the euchromatic arms for metacentric chromosomes (*Appendix 1—figure 3*) and contigs extended into the pericentromere in all cases which aided in the assembly of the pericentromeric region. Due to the fragmented Muller C in our EB assembly, we replaced the 22 Muller C EB contigs (above) with the scaffolded Muller C assembled from the EA (below) to create the 'Dath_EB_hybrid' genome (*Appendix 1—table 2*). Muller C EB SNPs were replaced with EA SNPs (see Materials and methods). This assembly had the highest genome completeness score of any of the *D. athabasca* assemblies (*Appendix 1—table 2*) and is what we present in the paper.

**Appendix 1—table 2.** Summary statistics and BUSCO results from genome assembly process of *Drosophila athabasca* EB

| | Canu + bandage | Canu + Quiver (2x) | Canu + Quiver (2x) + Pilon (2x) | Final Hi-C scaffolded Dath_EB_1.0 | Final Hi-C scaffolded Dath_EB_hybrid |
|---|---|---|---|---|---|
| N50 | 14,480,452 | 15,318,533 | 15,319,690 | * | * |
| Max Contig | 34,554,387 | 34,873,651 | 34,879,405 | * | * |
| Assembly Size | 195,713,124 | 192,740,469 | 192,655,157 | 192,660,667 | 192,054,219 |
| Number of Contigs | 199 | 133 | 133 | * | * |
| Complete BUSCOs | 1043 | 1054 | 1057 | 1057 | 1060 |
| Complete Single Copy BUS-COs | 1023 | 1046 | 1048 | 1048 | 1052 |
| Duplicated | 20 | 8 | 9 | 9 | 8 |
| Fragmented | 12 | 3 | 1 | 1 | 1 |
| Missing | 11 | 9 | 8 | 8 | 5 |
| % BUSCOs complete | 97.9% | 98.9% | 99.2% | 99.2% | 99.5% |

DOI: https://doi.org/10.7554/eLife.49002.043

## Dath EB

Muller A-AD

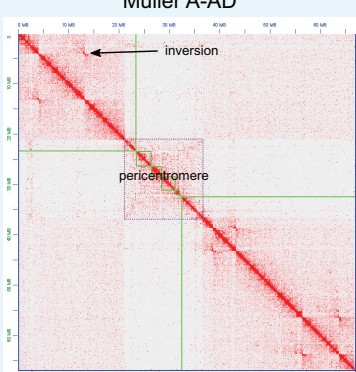

Muller B

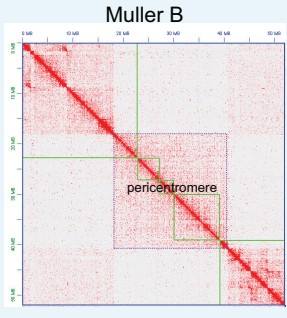

Muller E

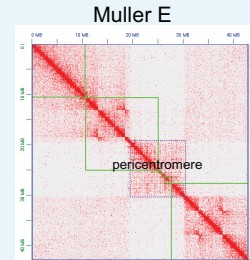

**Appendix 1—figure 3.** *Drosophila athabasca* EB metacentric chromosome Hi-C associations and scaffolding. Our EB assembly (*Appendix 1—table 2*) was superior to our EA assembly (*Appendix 1—table 3*) and long contigs from our EB assembly extended at least a megabase into the pericentromere for all metacentric chromosomes. Shown above are Hi-C association heatmaps from Juicebox (*Durand et al., 2016a*). Green boxes denote contigs. The pericentromeric region is highlighted in purple, and note the clear transition in Hi-C associations between euchromatic and heterochromatic regions. We used EA Hi-C data to scaffold the EB assembly. The EA and EB semispecies harbor inversions that differentiate the semispecies and we identified numerous inversions when mapping EA Hi-C to the EB genome assembly. Thus, the exceptionally long EB contigs that extend into the pericentromeric region allowed us to accurately scaffold chromosomes while simultaneously identifying inversions along the euchromatic arms.
DOI: https://doi.org/10.7554/eLife.49002.044

**EA assembly:** The initial canu assembly used the same parameters as used for the EB assembly (above and see Materials and methods). Our investigation of the resulting assembly using Bandage and BLAST identified 15.5 Mb of non-target sequence (primarily *Acetobacter* and *Sacchoramyces*) which was removed. Using Bandage we merged the unambiguous nodes 1334, 1335, 1336, 1337, 1338, 1340, 1342, and 1344 (total 11,326,223 bp), 1385 and 1387 (total 3,563,450 bp), 1369 and 162 (total 2,257,788 bp), 1328, 1330, and 1332 (total 5,082,211 bp). Assembly statistics for the EA assembly (*Appendix 1—table 3*) did not approach those of the EB assembly, likely do to the shorter raw PacBio reads that we had for this genome assembly (see Materials and methods). BUCSO statistics improved during genome polishing (*Appendix 1—table 3*) but were also slightly lower than our EB assembly (*Appendix 1—table 2*). Hi-C scaffolding was able to assemble the euchromatic arms and pericentromeric regions for all Muller elements (*Appendix 1—figure 4*). However, for Muller E, we found many uncollapsed haplotypes due to residual heterozygosity in the sequenced line. In addition, the overall shorter contig lengths failed to assemble across the euchromatic/heterochromatic boundary for some Muller elements, leaving the orientation of the euchromatic arm and the pericentromeric region more ambiguous (*Appendix 1—figure 4*).

**Appendix 1—table 3.** Summary statistics and BUSCO results from genome assembly process of *Drosophila athabasca* EA

|  | Canu + bandage | Canu + Quiver (2x) + Pilon (2x) | Final Hi-C scaffolded Dath_EA_1.0 |
|---|---|---|---|
| N50 | 5,537,664 | 5,538,275 | * |
| Max Contig | 20,031,442 | 20,052,631 | * |
| Assembly Size | 193,369,473 | 193,423,818 | 193,434,778 |

*Appendix 1—table 3 continued on next page*

Appendix 1—table 3 continued

|  | Canu + bandage | Canu + Quiver (2x) + Pilon (2x) | Final Hi-C scaffolded Dath_EA_1.0 |
|---|---|---|---|
| Number of Contigs | 348 | 348 | * |
| Complete BUSCOs | 1041 | 1055 | 1055 |
| Complete Single Copy BUSCOs | 1021 | 1041 | 1041 |
| Duplicated | 20 | 14 | 14 |
| Fragmented | 14 | 5 | 5 |
| Missing | 11 | 6 | 6 |
| % BUSCOs complete | 97.7% | 99.0% | 99.0% |

DOI: https://doi.org/10.7554/eLife.49002.045

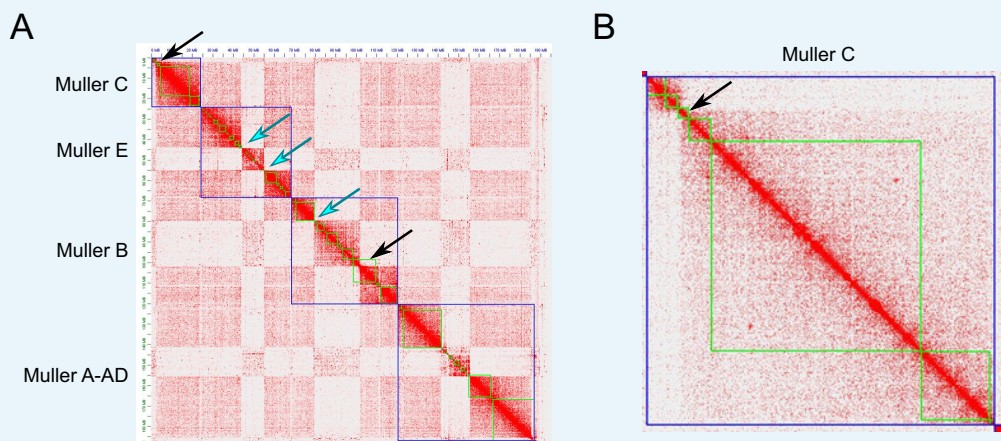

**Appendix 1—figure 4.** *Drosophila athabasca* EA assembly Hi-C associations and scaffolding. (**A**) Hi-C scaffolding of the EA assembly recovered long blocks of contigs we identified as Muller elements from our EB assembly. Blue boxes bound putative Muller element boundaries; green boxes denote contigs. Black arrows show contigs that span the euchromatic/heterochromatic transition and allow for confident scaffolding into pericentromeric regions. Blue arrows show regions where contigs failed to assemble across the transition making scaffolding based on Hi-C associations more challenging. (**B**) Shown is a zoomed in image of Muller C scaffolding. Here, a contig spans the euchromatic/ heterochromatic transition on Muller C and we used this scaffolded in our Dath_EB_hybrid assembly. Note the lack of evidence for inversions in the Hi-C heatmaps since here we are using EA Hi-C with an EA assembly.

DOI: https://doi.org/10.7554/eLife.49002.046

## Drosophila lowei

*Drosophila lowei* was the most fragmented of our assemblies (*Appendix 1—table 4*), probably because we sequenced an outbred strain that harbored high levels of heterozygosity (*D. lowei* is extremely difficult to maintain in the lab). Excessive heterozygosity is known to be challenging for assembly algorithms. The subsequent polishing steps using Racon and Pilon improved the assembly by reducing both the number of fragmented genes and missing genes and increasing the number of complete BUSCOs (*Appendix 1—table 4*). Our contig filtering process (see Materials and methods and above) identified multiple contigs with BLAST hits to *Acetobacter spp.* and *Saccharomyces cerevisiae* that also showed aberrant Illumina coverage patterns not consistent with our target species coverage (described above). These flagged contigs (amounting to a total of 7.1 Mb of sequence) were

removed from the assembly. Although the final assembly was more fragmented when compared to our other assemblies, Hi-C scaffolding oriented contigs into scaffolds that appeared highly accurate. For example, Muller B was assembled into a scaffold that was remarkably similar to Muller B of *D. pseudoobscura* (see RESULTS) and the Muller C pericentromeric region appeared largely syntenic across all obscura group species, including *D. lowei* (See RESULTS). In total, our results provide some evidence that for even our most fragmented assembly, we could correctly scaffold chromosomes using our Hi-C data. Hi-C scaffolding and contaminant removal during the final scaffolding step did lead to a small decrease in our BUSCO statistics (*Appendix 1—table 4*). Whole genome comparisons with close relative *D. miranda* helped confirm our scaffolding (*Appendix 1—figure 5*).

**Appendix 1—table 4.** Summary statistics and BUSCO results from genome assembly process of *Drosophila lowei*.

| | Canu/WTDBG2 only | Canu/WTDBG2 + Racon (3x) | Canu/WTDBG2 + Racon (3x) + Pilon (1x) | Canu/WTDBG2 + Racon (3x) + Pilon (2x) | Final Hi-C scaffolded Dlow_1.0 |
|---|---|---|---|---|---|
| N50 | 4,754,630 | 4,787,044 | 4,797,192 | 4,793,318 | * |
| Max Contig | 20,816,730 | 20,907,879 | 20,946,520 | 20,929,419 | * |
| Assembly Size | 192,748,718 | 191,586,436 | 191,816,863 | 191,620,915 | 184,313,494 |
| Number of Contigs | 943 | 726 | 726 | 726 | * |
| Complete BUSCOs | 936 | 975 | 1037 | 1039 | 1036 |
| Complete Single Copy BUS-COs | 920 | 960 | 1016 | 1021 | 1022 |
| Duplicated | 16 | 15 | 21 | 18 | 14 |
| Fragmented | 67 | 39 | 3 | 1 | 1 |
| Missing | 63 | 52 | 26 | 26 | 29 |
| % BUSCOs complete | 87.8% | 91.5% | 97.3% | 97.5% | 97.2% |

DOI: https://doi.org/10.7554/eLife.49002.047

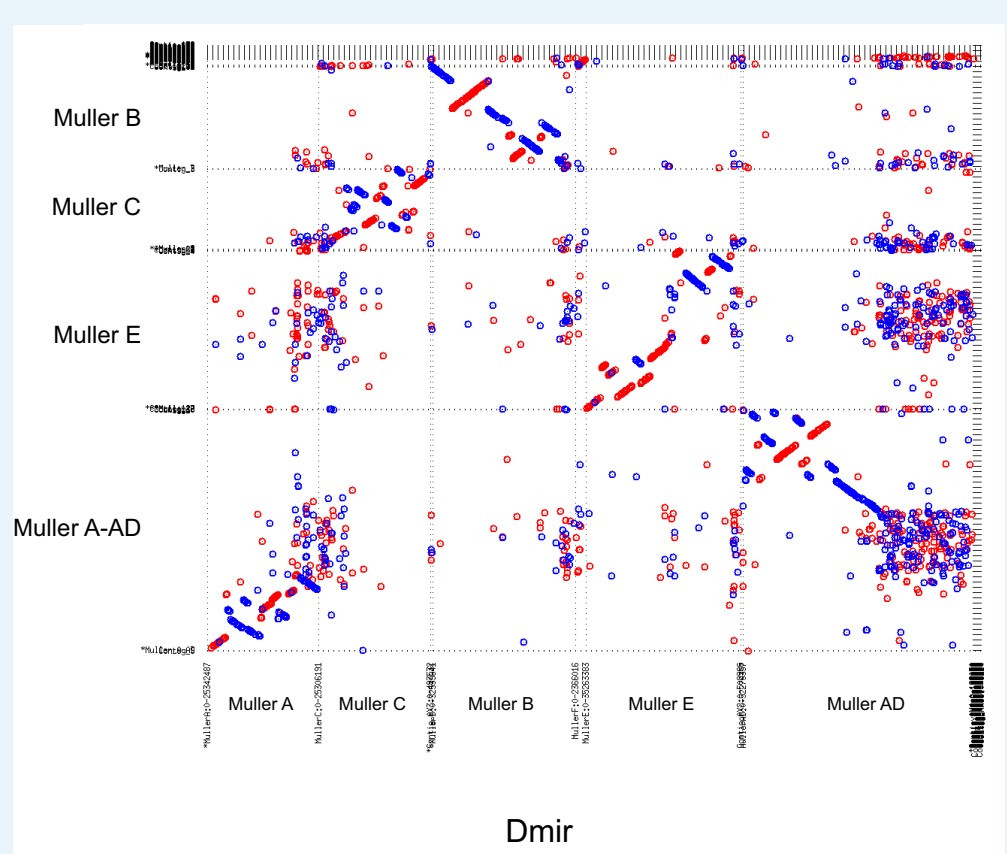

**Appendix 1—figure 5.** Whole genome alignment of our *D. lowei* assembly (Y axis) to the published *Drosophila miranda* genome.

DOI: https://doi.org/10.7554/eLife.49002.048

## Drosophila pseudoobscura

For *Drosophila pseudoobscua* we were interested in assembling the Muller elements and as much of the Y chromosome as possible. Therefore, we only Nanopore sequenced males. The availability of female Nanopore data of the same line (**Miller et al., 2018**) allowed us to increase our coverage for the autosomes and X chromosome by merging datasets. However, the varied coverage over each genomic partition (Autosomes >X >Y) led us to take an approach where we assembled each separately and then used Hi-C to scaffold the contigs together. We first targeted the autosomes for assembly using canu (see Materials and methods) and genomeSize = 200 m. All the resulting contigs >0.8 Mb that mapped to Muller elements B, C, E and F in *D. miranda* using MUMmer (**Kurtz et al., 2004**) were marked as autosomal. We then polished the whole assembly (Racon and Pilon) and then mapped the raw Nanopore reads to the assembly and subtracted out uniquely mapping autosomal reads. The remaining reads were deemed enriched for X or Y and assembled using canu (see Materials and methods) targeting the X chromosome with genomeSize = 100 m. We followed methods outlined above (i.e. mapping of resulting contigs to Muller A-AD from *D. miranda*) to assemble and then polished the assembly. Finally, using all Nanopore reads, we used the same subtraction method (above) and produced reads enriched for Y and unincorporated heterochromatin and performed a canu assembly with genomeSize = 50 m We then polished this final assembly using rounds of Racon and Pilon (**Appendix 1—table 5**). To finish, we again used Hi-C scaffolding to orient contigs from the entire assembly (autosomes, X and Y) while noting each contigs initial placement (above) and checking expected female/male Illumina coverage patterns for the autosomes, X chromosome, and Y to confirm assembly. Y chromosome contigs were not scaffolded.

**Appendix 1—table 5.** Summary statistics and BUSCO results from the genome assembly process of *Drosophila pseudoobscura*

| | Canu + Racon (3x) | Canu + Racon (3x) + Pilon (1x) | Canu + Racon (3x) + Pilon (2x) | Final Hi-C scaffolded Dpse_1.0 |
|---|---|---|---|---|
| N50 | 5,996,964 | 5,975,358 | 5,971,646 | * |
| Max Contig | 20,387,169 | 20,334,965 | 20,319,488 | * |
| Assembly Size | 194,744,540 | 194,172,171 | 193,935,436 | 193,980,066 |
| Number of Contigs | 481 | 481 | 481 | * |
| Complete BUSCOs | 974 | 1052 | 1057 | 1055 |
| Complete Single Copy BUSCOs | 968 | 1043 | 1048 | 1048 |
| Duplicated | 6 | 9 | 9 | 7 |
| Fragmented | 59 | 3 | 2 | 3 |
| Missing | 33 | 11 | 7 | 8 |
| % BUSCOs complete | 91.4% | 98.6% | 99.2% | 99.0% |

DOI: https://doi.org/10.7554/eLife.49002.049

We compared our genome assembly with the Dpse_r3.04 and found widespread concordance (*Appendix 1—figure 6*). However, we see an improvement over the published assembly with regards to Muller B and Muller A-AD and there is a substantial increase in the amount of pericentromeric sequence assigned to all Muller elements in our assembly (*Appendix 1—figure 6*). BUSCO statistics were found to be rather high and comparable to our other genome assemblies.

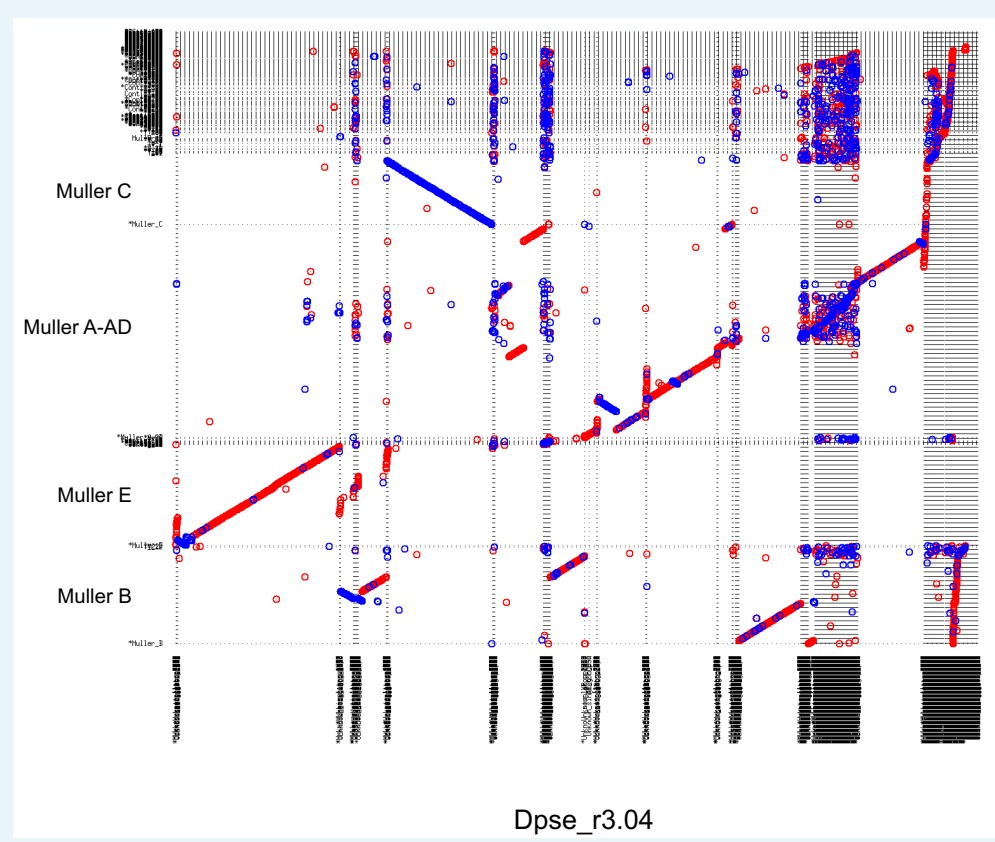

**Appendix 1—figure 6.** Whole genome alignment of our assembly (Y axis) to the published *Drosophila pseudoobscura* genome assembly (version 3.04). Scaffolds in the published assembly that are near chromosome length (i.e., Muller E and Muller C) largely agree with our scaffolds. However, our assembly extends the assembled length of these chromosomes with far less scaffolding. For Muller B and Muller A-AD, our scaffolded chromosomes show large stretches of collinearity with the fragmented published assembly, with the exception of a few inverted regions. Our Hi-C data and association heatmap (see RESULTS) argue that our assembly orientation is likely the correct one and provide orientation to the five large scaffolds of Muller B and 8 scaffolds of Muller A-AD in the reference genome. The paleocentromeric region on Muller E is assembled in the current reference genome, but our assembly contains additional sequence not present in the published assembly (not shown).
DOI: https://doi.org/10.7554/eLife.49002.050

