## [Decision Letter]

Thank you for submitting your article "Dynamic turnover of centromeres drives karyotype evolution in *Drosophila*" for consideration by *eLife*. Your article has been reviewed by three peer reviewers, including Steven Henikoff as the Reviewing Editor and Reviewer #1, and the evaluation has been overseen by Diethard Tautz as the Senior Editor.

The reviewers have discussed the reviews with one another and the Reviewing Editor has drafted this decision to help you prepare a revised submission.

Summary:

The manuscript describes assemblies of genomic sequence from *Drosophila* species in the obscura group to trace the evolution of centromeres and pericentromeres. Unexpectedly, they find de novo centromere formation rather than chromosome inversion to be the common mode of centromere re-location, and that new centromeres can form in gene-rich regions with higher AT content. Repetitive sequences rapidly accumulate around new centromeres and are rapidly lost when the centromere changes position. This is a comprehensive study of centromere and karyotype evolution that will be of interest to investigators of centromeres and heterochromatin and to evolutionary biologists. The genome assemblies will facilitate other evolutionary studies in *Drosophila*.

Essential revisions:

The reviewers all agree that the long-read assembly of genomes in the *D. obscura* group is comprehensive and of high technical quality, and that the study addresses an interesting question in chromosome evolutionary dynamics in a well-studied model clade. No further experimental data are necessary, however additional analyses will be required for publication.

1) Mapping centromeric regions using H3K9me3 ChIP-seq cannot resolve centromeres from peri-centromeres and leaves large uncertainties, and analysis of repeat content in unplaced scaffolds, for example by k-mer analysis, will help fill in the blanks.

2) The manuscript should be edited to reflect the limitations of the data as presented and include clarifications of the centromeric and pericentric differences to the extent that they can be assessed.

---

## [Author Response]

Essential revisions:

*The reviewers all agree that the long-read assembly of genomes in the* D. obscura *group is comprehensive and of high technical quality, and that the study addresses an interesting question in chromosome evolutionary dynamics in a well-studied model clade. No further experimental data are necessary, however additional analyses will be required for publication.*

We thank the reviewers for the careful reading of our manuscript. We have incorporated all of the requested changes.

1) Mapping centromeric regions using H3K9me3 ChIP-seq cannot resolve centromeres from peri-centromeres and leaves large uncertainties, and analysis of repeat content in unplaced scaffolds, for example by k-mer analysis, will help fill in the blanks.

This is a good point. We now include new analyses where we identify tandem repeats directly from raw sequencing reads, independent of an assembly. We took advantage of a new algorithm (TideHunter) for detecting tandem repeats directly from long-read sequencing data and include this analysis in our Results section (subsection “Identification of centromere-associated repeats”, new Figure 4—figure supplement 2, Table 2). Further, we include more information on our k-Seek analysis in the paper, which detects satellites from Illumina sequences and is also independent of the assembly. In our previous version we only discussed the k-Seek analysis in *D. subobscura*, and we now also incorporate and discuss the results of this analysis in species other than *D. subobscura*. Importantly, the bioinformatic results largely agree with each other (Table 2). We clarify in the paper that these three bioinformatic approaches (Tandem Repeat Finder, k-Seek, and TideHunter), along with FISH for specific satellites, gives us added certainty about the centromeric associated sequences.

2) The manuscript should be edited to reflect the limitations of the data as presented and include clarifications of the centromeric and pericentric differences to the extent that they can be assessed.

We agree. We make sure that we emphasize throughout the manuscript that we cannot distinguish centromeric regions from the pericentromere. In particular, we modify the wording in the Abstract, Introduction, and throughout the Results section, and in the Discussion. We clarify that the ‘centromere repeats’ are ‘centromere-associated repeats’ throughout the manuscript, and that these centromere-associated satellites may not form the functional centromere. We add a new paragraph in the Results section:

“Thus, our bioinformatics analysis identifies putative centromere-associated repeats in each species, and FISH supports their location near the centromere. Importantly, however, our experiments do not allow us to identify the functional centromere. Indeed, initial FISH experiments in *D. melanogaster* suggested that the functional centromere in that species consist of simple satellites (citations). However, recent high-resolution chromatin fiber imaging and mapping of the centromere protein instead indicate that island of retroelements, flanked by large arrays of satellites are the major functional components of centromeres in *D. melanogaster* (Talbert et al., 2018; Chang et al., 2019). It will be of great interest to perform similar mapping studies in the obscura group, to determine the functional centromere and its evolution in this species group.“

Thus, we clearly spell out that future experiments are necessary to identify the functional centromere in flies of the *obscura* group.